# Somatostatin binds to the human amyloid β peptide and favors the formation of distinct oligomers

Hansen Wang[1], Lisa D Muiznieks[2], Punam Ghosh[3], Declan Williams[1], Michael Solarski[1,4], Andrew Fang[5,6], Alejandro Ruiz-Riquelme[1], Régis Pomès[2,7], Joel C Watts[1,7], Avi Chakrabartty[3,7], Holger Wille[5,6], Simon Sharpe[2,7], Gerold Schmitt-Ulms[1,4]*

[1]Tanz Centre for Research in Neurodegenerative Diseases, University of Toronto, Toronto, Canada; [2]Molecular Medicine Program, Research Institute, The Hospital for Sick Children, Toronto, Canada; [3]Department of Medical Biophysics, University of Toronto, Toronto, Canada; [4]Department of Laboratory Medicine and Pathobiology, University of Toronto, Toronto, Canada; [5]Department of Biochemistry, University of Alberta, Edmonton, Canada; [6]Centre for Prions and Protein Folding Diseases, University of Alberta, Edmonton, Canada; [7]Department of Biochemistry, University of Toronto, Toronto, Canada

**Abstract** The amyloid β peptide (Aβ) is a key player in the etiology of Alzheimer disease (AD), yet a systematic investigation of its molecular interactions has not been reported. Here we identified by quantitative mass spectrometry proteins in human brain extract that bind to oligomeric Aβ1-42 (oAβ1-42) and/or monomeric Aβ1-42 (mAβ1-42) baits. Remarkably, the cyclic neuroendocrine peptide somatostatin-14 (SST14) was observed to be the most selectively enriched oAβ1-42 binder. The binding interface comprises a central tryptophan within SST14 and the N-terminus of Aβ1-42. The presence of SST14 inhibited Aβ aggregation and masked the ability of several antibodies to detect Aβ. Notably, Aβ1-42, but not Aβ1-40, formed in the presence of SST14 oligomeric assemblies of 50 to 60 kDa that were visualized by gel electrophoresis, nanoparticle tracking analysis and electron microscopy. These findings may be relevant for Aβ-directed diagnostics and may signify a role of SST14 in the etiology of AD.

*For correspondence:
g.schmittulms@utoronto.ca

## Introduction

The amyloid beta (Aβ) peptide represents a 37 to 49 amino acids endoproteolytic fragment of the amyloid precursor protein (APP), a single-span transmembrane protein, which in humans is coded on the long arm of Chromosome 21 within cytogenetic band q21.3 (*Kang et al., 1987*). The cellular biology that governs the formation and clearance of Aβ has been understood to play a critical role in Alzheimer disease (AD) since it was shown that fibrillary aggregates of Aβ represent the main constituents of extracellular amyloid plaques that accumulate in the brains of individuals afflicted with the disease (*Glenner and Wong, 1984*). The endoproteinases responsible for the release of Aβ are broadly referred to as secretases, with β- and γ-secretases being responsible for the critical cleavage reactions that cause the N- and C-terminal release of the Aβ peptide from its APP precursor, respectively. Mutations in the human APP gene (*Goate et al., 1991*) or the genes coding for presenilin 1 or 2, catalytic subunits of γ-secretases (*Rogaev et al., 1995*; *Sherrington et al., 1995*), which cause an increase in Aβ levels or shift the balance of Aβ peptides of different lengths in favor of the production of Aβ1-42, remain the only known causes for early-onset familial manifestations of AD.

**eLife digest** Treating Alzheimer's disease and related dementias is one of the major challenges currently facing healthcare providers worldwide. A hallmark of the disease is the formation of large deposits of a specific molecule, known as amyloid beta (Aβ), in the brain. However, more and more research suggests that smaller and particularly toxic amyloid beta clumps – often referred to as oligomeric Aβ – appear as an early sign of Alzheimer's disease.

To understand how the formation of these smaller amyloid beta clumps triggers other aspects of the disease, it is important to identify molecules in the human brain that oligomeric Aβ binds to. To this end, Wang et al. attached amyloid beta or oligomeric Aβ molecules to microscopically small beads. The beads were then exposed to human brain extracts in a test tube, which allowed molecules in the extracts to bind to the amyloid beta or oligomeric Aβ. The samples were then spun at high speed, meaning that the beads and any other molecules bound to them sunk and formed pellets at the bottom of the tubes. Each pellet was then analyzed to see which molecules it contained.

The experiments identified more than a hundred human brain proteins that can bind to amyloid beta. One of them, known as somatostatin, selectively binds to oligomeric Aβ. Wang et al. were able to determine the structural features of somatostatin that control this binding.

Finally, in further experiments performed in test tubes, Wang et al. noticed that smaller oligomeric Aβ clumps were more likely to form than larger amyloid beta deposits when somatostatin was present. This could signify a previously unrecognized role of somatostatin in the development of Alzheimer's disease.

Further studies are now needed to confirm whether the presence of somatostatin in the brain favors the formation of smaller, toxic oligomeric Aβ clumps over large innocuous amyloid beta deposits. If so, new treatments could be developed that aim to reduce oligomeric Aβ levels in the brain by preventing somatostatin from interacting with amyloid beta molecules. Wang et al. also suggest that somatostatin could be used in diagnostic tests to detect abnormal levels of oligomeric Aβ in the brain or body fluids of people who have Alzheimer's disease.

Although the extracellular amyloid deposits themselves are increasingly being viewed as innocuous sinks for misfolded Aβ, they may continue to play a role as reservoirs from which monomeric or oligomeric Aβ, hereafter referred to as mAβ or oAβ (see also (*Yang et al., 2017*)), can emanate (*Selkoe, 2001*). These soluble forms of Aβ may interact with other molecules they meet in the extracellular space (*Narayan et al., 2011*), or may bind to molecules embedded in the plasma membrane (*Laurén et al., 2009*), which may be a precursor to their endocytosis (*Jin et al., 2016*). Although it is not known how precisely Aβ, which has been taken up into the cell by endocytosis, can overcome the bilayer that surrounds the endolysosomal vesiculo-tubular network, it is well established that the peptide can also reach the cytoplasm and associates with other intracellular compartments, including mitochondria (*Lustbader et al., 2004*).

The ability of Aβ to overcome compartmental boundaries allows the peptide to encounter a wide variety of proteins that may essentially exist anywhere in the brain. Within the extracellular space, Aβ has, for instance, been shown to interact with apolipoproteins E (*Strittmatter et al., 1993*) and J (the latter is also known as clusterin) (*Narayan et al., 2011*; *Ghiso et al., 1993*). These interactions with apolipoproteins are relevant in the AD context, as certain polymorphisms in the genes coding for them have been shown to bestow an increased risk to develop late-onset AD (*Lambert et al., 2009*; *Harold et al., 2009*; *Corder et al., 1993*). When neurons are exposed to soluble oAβ, a cascade of events is thought to unfold, which leads to the intracellular deposition of hyperphosphorylated tau in the form of so-called neurofibrillary tangles (NFT) (*Grundke-Iqbal et al., 1986a*, *1986b*). The relationship between these two pathobiological features of the disease has remained enigmatic and, although there is broad agreement that signaling downstream of oAβ is toxic for cells, a bewildering number of hypotheses exist regarding the mechanism by which oAβ-dependent toxicity manifests. Receptor candidates proposed to mediate oAβ toxicity include RAGE receptors (*Origlia et al., 2008*), insulin receptor-sensitive Aβ-binding protein (*De Felice et al., 2009*), P/Q

calcium channels (*Nimmrich et al., 2008*), sphingomyelinase (*Grimm et al., 2005*), the Aβ parent molecule APP itself (*Lorenzo et al., 2000*; *Shaked et al., 2006*; *Sola Vigo et al., 2009*), amylin receptors (*Fu et al., 2012*), a subset of integrins (*Wright et al., 2007*), the prion protein (PrP) (*Laurén et al., 2009*) and the metabotropic glutamate receptor 5 (which may act as a co-receptor for PrP (*Um et al., 2013*; *Hu et al., 2014*)). Within cells, Aβ has been reported to associate with more than a dozen additional proteins, including glyceraldehyde dehydrogenase (GAPDH) (*Verdier et al., 2005*), the mitochondrial ATP synthase complex (*Schmidt et al., 2008*), and a 17-β-hydroxysteroid dehydrogenase X (HSD10) (*Yan et al., 1997*), also known as 3-hydroxyacyl-CoA dehydrogenase type-2 or Aβ-binding alcohol dehydrogenase (ABAD).

Countless isolated experimental paradigms, differences in Aβ preparations and a broad spectrum of methods underlie the discoveries of the aforementioned Aβ binding candidates, a dissatisfactory status quo, also lamented by others (*Mucke and Selkoe, 2012*). We therefore set out to locate pertinent literature studies that made use of a more systematic mass spectrometry (MS)-based discovery approach for the identification of Aβ interactors. Surprisingly, whereas several studies are available that interrogated the APP interactome by affinity purification MS (*Kohli et al., 2012*; *Bai et al., 2008*), a similar investigation has, to our knowledge, not yet been reported for the Aβ peptide.

To address this shortcoming, we have now undertaken an in-depth search for proteins that can bind to Aβ using biotinylated Aβ peptides as baits and human frontal lobe extract as the biological source for the capture of Aβ binding proteins. We made use of isobaric tagging for the relative quantitation of proteins and capitalized on recent advances in mass spectrometry instrumentation. In addition to confirming many of the previously known Aβ interactors and revealing more than a hundred novel Aβ binding candidates, the study uncovered a surprising and selective interaction between oligomeric Aβ and the small cyclic peptide somatostatin (SST). We demonstrate that SST (i) preferentially binds to aggregated Aβ, (ii) influences Aβ aggregation, (iii) traps a proportion of Aβ in oligomeric assemblies of 50 to 60 kDa, (iv) masks the ability of several widely-used antibodies to detect Aβ, and (v) may form a complex with monomeric Aβ that can induce tau hyperphosphorylation in primary hippocampal neurons.

## Results

### Workflow of Aβ interactome analyses

The primary objective of this study was to generate an in-depth inventory of human brain proteins that oligomeric preparations of Aβ1-42 can bind to using an unbiased in vitro discovery approach. Synthetic Aβ1-42 peptides and brain extracts generated from adult human frontal lobe tissue served in these studies as baits and biological source materials, respectively. oAβ1-42 was prepared by aggregating the peptide at 4°C for 24 hr, using previously described procedures known to generate amyloid-β-derived diffusible ligands (ADDLs) (*De Felice et al., 2009*; *Krafft and Klein, 2010*). ADDLs are understood to be composed of a heterogenous mixture of oligomeric and prefibrillar Aβ aggregates. This heterogeneity ensured that the analysis was not limited to particular oligomeric Aβ assemblies, which were observed to predominate with some alternative preparation protocols (*Barghorn et al., 2005*; *Ahmed et al., 2010*). Because the interaction with a given binding partner may involve a binding epitope that comprises N- or C-terminal residues of Aβ1-42, initially two separate experiments (I and II) were conducted, which differed in the orientation designated for tethering the oAβ1-42 bait to the affinity matrix (*Figure 1A*). To facilitate meaningful comparisons across experiments, the method of Aβ1-42 capture was not based on immunoaffinity reagents. Instead, alternative Aβ1-42 baits were equipped with biotin moieties attached to the N- or C-terminus by a 6-carbon linker chain, enabling their consistent affinity-capture on streptavidin agarose matrices. Large aggregates were removed prior to the bait capture step by centrifugal sedimentation. Biotin-saturated streptavidin agarose matrices served as negative controls and three biological replicates of samples and controls were generated for each interactome dataset by reproducing the affinity-capture step side-by-side on three separate streptavidin agarose affinity matrices that had been saturated with the biotinylated baits. To identify differences in protein-protein interactions of monomeric versus oligomeric Aβ1-42, a third interactome experiment was conducted in which oAβ1-42-biotin or mAβ1-42-biotin served as baits. Digitonin-solubilized brain extracts, which are known to primarily comprise extracellular and cellular proteins (except for nuclear proteins) served as

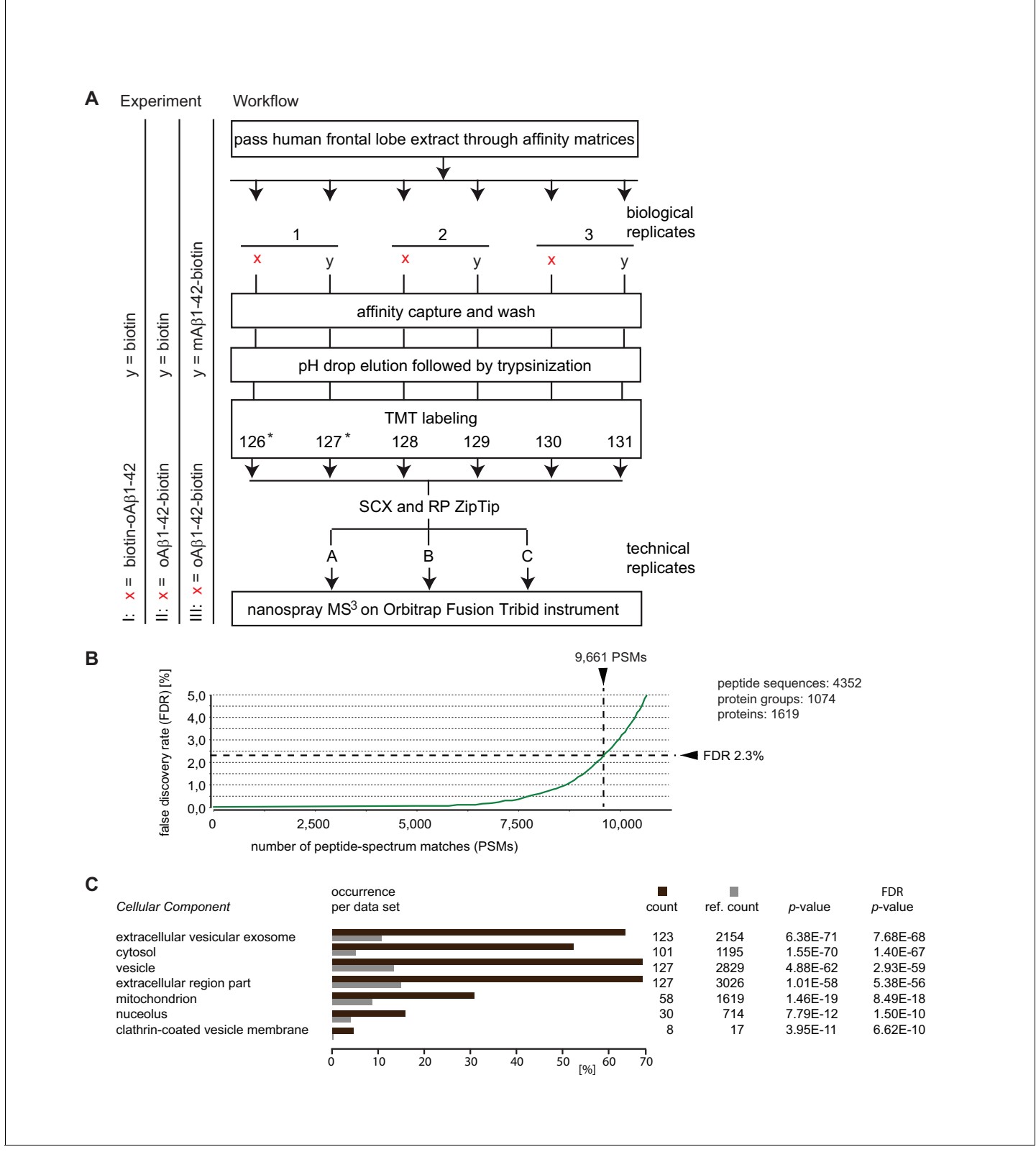

**Figure 1.** Summary of Aβ1-42 interactome analyses. (**A**) Workflow of interactome studies designed to capture binders to oligomeric Aβ1-42 tethered to the streptavidin matrix by N-terminal (Experiment I) or C-terminal (Experiment II) biotin groups, or comparing binders to oligomeric versus monomeric Aβ1-42 (Experiment III). (**B**) Representative chart from interactome dataset generated in Experiment I, depicting the false discovery rates of peptide-to-spectrum matches and benchmarks of the analysis depth. (**C**) 'Cellular Component' Gene Ontology analysis of top 200 proteins that exhibited the most

*Figure 1 continued on next page*

*Figure 1 continued*

pronounced oAβ1-42 co-enrichment in Experiment I on the basis of their isobaric signature ion distribution. Asterisks indicate TMT labels that were omitted in a subset of quantitative mass spectrometry experiments.

The following source data is available for figure 1:

**Source data 1.** Experiment I-III interactome data (alphabetically sorted) (Excel file).

biological starting materials, consistent with the main subcellular areas previously reported to harbor Aβ. Following extensive washes of affinity matrices in their protein-bound state, binders to the bait peptides were eluted by rapid acidification, fully denatured in 9 M urea, and trypsinized. To avoid notorious confounders related to variances in the subsequent handling and analysis of samples, individual peptide mixtures were labeled with distinct isobaric tandem mass tags (TMT) in a six-plex format, then combined and concomitantly subjected to ZipTip-based pre-analysis clean-up by strong cation exchange (SCX) and reversed phase (RP) separation. Four-hour split-free reversed phase nanospray separations were online coupled to an Orbitrap Fusion Tribrid mass spectrometer, which was configured to run an MS3 analysis method. Tandem MS spectra were matched to peptide sequences by interrogating the human international protein index (IPI) using Sequest and Mascot algorithms. The relative levels of individual peptides in the six samples could be determined by comparing the intensity ratios of the corresponding TMT signature ions in the low mass range of MS$^3$ fragment spectra.

## The Aβ1-42 interactome

The three comparative Aβ1-42 interactome analyses (Experiments I-III) conducted in this study led to mass spectrometry datasets, which were characterized by similar benchmarks of data quality and enabled confident assignments of several thousand mass spectra to human peptides. For example, Experiment I generated 9661 spectra, which passed confidence criteria applied. These filtered spectra could be matched to 4352 unique peptides, which in turn formed the basis for the identification of 1074 protein groups. The designation 'protein groups', as opposed to 'proteins', reflects a reality of a subset of tryptic peptide sequences not being uniquely associated with a specific protein. Whenever encountered, only ambiguous assignments are possible. For this specific dataset, the 1074 protein groups were annotated to comprise 1619 unique proteins. No attempt was made to resolve this residual source of ambiguity at the individual peptide level. Instead, a majority of uncertain identifications were removed by requiring protein identifications to be based on the confident assignment of at least three unique peptides with a minimum length of six amino acids (*Figure 1B*).

To begin to characterize the Aβ1-42 interactome, the top-listed 200 proteins, whose levels were most pronouncedly co-enriched with biotin-oAβ1-42, were subjected to a gene ontology (GO) analysis. This analysis revealed amongst the biotin-oAβ1-42 binding candidates a highly significant overrepresentation (p<1.0 E-50) of 'Cellular component' GO classifiers that indicate an association with the 'extracellular vesicular exosome', 'cytosol', 'vesicle' and 'extracellular region part' (*Figure 1C*). A more detailed analysis of the list of biotin-oAβ1-42 (B-oAβ1-42) candidate interactors revealed that all top-listed 50 candidates were identified on the basis of more than ten peptide-to-spectrum matches (PSMs) and with sequence coverages exceeding 50% of their primary structures (*Table 1*). Moreover, indicative of high selectivity of the affinity capture procedures applied, the abundance levels of these 50 candidate interactors in the B-oAβ1-42-specific samples exceeded their levels in the negative control sample by more than four-fold. Not surprisingly, the Aβ peptide itself was amongst the proteins identified whose enrichment levels in specific versus unspecific affinity capture samples were most pronounced. In contrast, assignments of APP peptides that map to regions outside of Aβ were low scoring (that is, did not pass a 95% significance threshold) and therefore not credible. The highest levels of B-oAβ1-42 co-enrichment were exhibited by peroxiredoxin-5, rab-1A, lactate dehydrogenase, hypoxanthine-guanine phosphoribosyltransferase, ATP synthase subunits beta and gamma, and acyl CoA thioester hydrolase. Several of the Aβ1-42-biotin candidate interactors revealed by this analysis were previously known to interact with Aβ peptides, including ATP synthase (*Verdier et al., 2005*; *Schmidt et al., 2008*) and glyceraldehyde-3-phosphate dehydrogenase (*Verdier et al., 2005*, *2008*), and/or had been shown to be present at altered abundance levels in

**Table 1.** Top-listed 50 interactors of biotin-Aβ1-42 observed in Experiment I. Except for the Aβ1-42 bait, which is shown in first position, proteins are listed by their enrichment (relative to the biotin-only saturated negative control matrix) observed in Experiment I. Note the extensive amino acid sequence coverage exceeding 44% for all proteins listed. Whenever the same proteins were also observed in Experiments II and III, their corresponding enrichment ratios and counts of peptides quantified are shown in additional columns on the right. (see also **Figure 1—source data 1** and **Supplementary file 1**).

| Accession | Description | Coverage | Peptides Unique | Total | Experiment I B-oAβ1-42/B Ratio | Count | Experiment II oAβ1-42-B/B Ratio | Count | *Experiment III oAβ1-42-B/ mAβ1-42-B Ratio | Count | AAs |
|---|---|---|---|---|---|---|---|---|---|---|---|
| IPI00006608.1 | APP770 of Amyloid beta A4 protein (Fragment) | 62.34% | 2 | 48 | 8.33 | 18 | 5.77 | 13 | 2.80 | 37 | 770 |
| IPI00024915.3 | Peroxiredoxin-5, mitochondrial | 86.45% | 8 | 18 | 12.23 | 3 | 2.15 | 3 | 5.90 | 23 | 214 |
| IPI00005719.1 | Ras-related protein Rab-1A | 80.49% | 4 | 18 | 10.52 | 3 | | | 3.28 | 6 | 205 |
| IPI00219217.3 | L-lactate dehydrogenase B chain | 74.55% | 6 | 26 | 9.87 | 9 | | | 2.46 | 3 | 334 |
| IPI00218493.7 | Hypoxanthine-guanine phosphoribosyltransferase | 64.22% | 7 | 22 | 9.00 | 12 | | | 2.45 | 8 | 218 |
| IPI00303476.1 | ATP synthase subunit beta, mitochondrial | 83.55% | 14 | 48 | 8.59 | 3 | | | 2.61 | 21 | 529 |
| IPI00395769.2 | ATP synthase subunit gamma, mitochondrial | 59.60% | 6 | 25 | 8.57 | 3 | | | 3.29 | 19 | 297 |
| IPI00219452.1 | Cytosolic acyl CoA thioester hydrolase | 76.60% | 12 | 31 | 8.38 | 31 | | | 4.03 | 31 | 329 |
| IPI00027223.2 | Isocitrate dehydrogenase [NADP] cytoplasmic | 62.56% | 5 | 30 | 8.20 | 4 | | | | | 414 |
| IPI00743713.4 | Pyruvate kinase | 64.77% | 3 | 41 | 7.95 | 5 | 1.89 | 7 | 4.50 | 8 | 599 |
| IPI00029469.1 | Centractin, beta | 76.06% | 3 | 26 | 7.87 | 7 | | | 3.25 | 39 | 376 |
| IPI00003482.1 | 2,4-dienoyl-CoA reductase, mitochondrial | 88.06% | 12 | 37 | 7.74 | 21 | | | 4.32 | 28 | 335 |
| IPI00000949.1 | Mu-crystallin homolog | 80.25% | 11 | 22 | 7.65 | 11 | | | 2.41 | 17 | 314 |
| IPI00028520.2 | NADH dehydrogenase flavoprotein 1 | 77.59% | 12 | 31 | 7.52 | 32 | 3.48 | 4 | 4.08 | 20 | 464 |
| IPI00018352.1 | Ubiquitin carboxyl-terminal hydrolase L1 | 63.68% | 9 | 17 | 7.49 | 38 | 1.87 | 2 | 1.39 | 19 | 223 |
| IPI00478231.2 | Transforming protein RhoA | 51.30% | 5 | 11 | 7.48 | 7 | | | 5.74 | 8 | 193 |
| IPI00028091.3 | Actin-related protein 3 | 54.78% | 6 | 24 | 7.45 | 9 | | | 1.81 | 4 | 418 |
| IPI00029468.1 | Centractin, alpha | 72.34% | 7 | 36 | 7.34 | 15 | 3.94 | 3 | 3.11 | 38 | 376 |
| IPI00335509.3 | Dihydropyrimidinase-related protein 5 | 77.13% | 8 | 42 | 7.05 | 6 | | | 3.79 | 8 | 564 |
| IPI00479186.7 | Pyruvate kinase isozymes M1/M2 | 93.03% | 3 | 70 | 7.04 | 138 | 1.68 | 100 | 4.33 | 123 | 531 |
| IPI00219018.7 | Glyceraldehyde-3-phosphate dehydrogenase | 98.21% | 35 | 50 | 6.96 | 160 | 2.36 | 121 | 2.62 | 251 | 335 |
| IPI00873027.2 | Carbohydrate kinase domain-containing protein | 95.68% | 15 | 31 | 6.95 | 18 | 2.17 | 3 | 1.72 | 32 | 347 |
| IPI00179187.4 | DnaJ homolog subfamily A member 3 | 74.17% | 6 | 31 | 6.93 | 3 | | | 9.57 | 8 | 453 |
| IPI00216456.5 | Histone H2A type 1 C | 92.31% | 3 | 21 | 6.83 | 15 | 2.03 | 10 | 0.59 | 54 | 130 |
| IPI00016801.1 | Glutamate dehydrogenase 1, mitochondrial | 85.13% | 13 | 46 | 6.82 | 16 | | | 2.99 | 22 | 558 |
| IPI00218448.4 | Histone H2A.Z | 83.59% | 5 | 14 | 6.78 | 4 | 1.88 | 5 | 0.51 | 30 | 128 |
| IPI00220301.5 | Peroxiredoxin-6 | 80.80% | 12 | 18 | 6.69 | 7 | 1.42 | 2 | 1.54 | 22 | 224 |
| IPI00472047.1 | NAD-dependent deacetylase sirtuin-2 | 78.32% | 7 | 28 | 6.68 | 4 | | | 2.58 | 15 | 369 |

*Table 1 continued on next page*

*Table 1 continued*

| Accession | Description | Coverage | Peptides Unique | Total | Experiment I B-oAβ1-42/B Ratio | Count | Experiment II oAβ1-42-B/B Ratio | Count | *Experiment III oAβ1-42-B/ mAβ1-42-B Ratio | Count | AAs |
|---|---|---|---|---|---|---|---|---|---|---|---|
| IPI00169383.3 | Phosphoglycerate kinase 1 | 92.09% | 18 | 51 | 6.61 | 48 | 1.38 | 13 | 1.73 | 21 | 417 |
| IPI00895801.1 | Medium-chain specific acyl-CoA dehydrogenase | 69.41% | 8 | 39 | 6.39 | 3 | | | 1.82 | 12 | 425 |
| IPI00026268.3 | Guanine nucleotide-binding protein beta-1 | 83.82% | 7 | 35 | 6.38 | 4 | | | 2.58 | 30 | 340 |
| IPI01015522.1 | Actin, cytoplasmic 1 | 96.83% | 8 | 37 | 6.25 | 33 | 2.07 | 26 | 4.71 | 129 | 347 |
| IPI00909207.1 | Peroxiredoxin-2 | 92.35% | 9 | 23 | 6.00 | 13 | 1.51 | 14 | 2.40 | 13 | 183 |
| IPI00970967.1 | GSTT2 protein | 51.64% | 5 | 14 | 6.00 | 7 | | | | | 244 |
| IPI00298547.3 | DJ-1 | 95.24% | 11 | 22 | 5.87 | 17 | 1.42 | 4 | 1.84 | 30 | 189 |
| IPI00796462.1 | GTP-binding nuclear protein Ran | 90.60% | 13 | 24 | 5.82 | 26 | 4.82 | 14 | 4.42 | 46 | 234 |
| IPI00784154.1 | Heat shock protein, 60 kDa, mitochondrial | 94.76% | 35 | 71 | 5.68 | 69 | 2.27 | 17 | 4.21 | 73 | 573 |
| IPI00003362.3 | 78 kDa glucose-regulated protein | 58.72% | 6 | 44 | 5.65 | 20 | 1.39 | 1 | 2.42 | 7 | 654 |
| IPI00011229.1 | Cathepsin D | 84.95% | 11 | 35 | 5.52 | 15 | | | 2.10 | 26 | 412 |
| IPI00013683.2 | Tubulin beta-3 chain | 84.44% | 10 | 43 | 5.39 | 5 | 1.18 | 1 | 2.04 | 85 | 450 |
| IPI00032406.1 | DnaJ homolog subfamily A member 2 | 90.29% | 10 | 38 | 5.28 | 5 | | | 6.10 | 19 | 412 |
| IPI00419802.4 | 3-hydroxyisobutyryl-CoA hydrolase | 64.25% | 5 | 22 | 5.27 | 7 | | | | | 386 |
| IPI00219037.5 | Histone H2A.x | 92.31% | 5 | 23 | 5.23 | 3 | 1.69 | 9 | 0.67 | 48 | 143 |
| IPI00556376.2 | Dihydropyrimidinase-related protein 1 | 89.21% | 21 | 60 | 5.21 | 57 | 1.39 | 27 | 3.71 | 67 | 686 |
| IPI00000792.1 | Quinone oxidoreductase | 68.69% | 3 | 18 | 5.18 | 4 | | | | | 329 |
| IPI00295400.1 | Tryptophanyl-tRNA synthetase, cytoplasmic | 66.67% | 6 | 33 | 5.17 | 9 | | | 3.97 | 2 | 471 |
| IPI00440493.2 | ATP synthase subunit alpha, mitochondrial | 99.46% | 38 | 89 | 5.17 | 62 | 2.16 | 32 | 4.88 | 131 | 553 |
| IPI00029111.3 | Dihydropyrimidinase-related protein 3 | 80.99% | 4 | 49 | 5.13 | 15 | 1.42 | 10 | 3.40 | 19 | 684 |
| IPI00019171.1 | Endophilin-A1 | 78.41% | 10 | 37 | 5.03 | 7 | 1.28 | 1 | 1.07 | 40 | 352 |
| IPI00005198.2 | Interleukin enhancer-binding factor 2 | 44.36% | 3 | 13 | 4.98 | 4 | | | 1.41 | 6 | 390 |
| IPI00030363.1 | Acetyl-CoA acetyltransferase, mitochondrial | 86.65% | 19 | 50 | 4.81 | 33 | 2.64 | 10 | 2.45 | 43 | 427 |
| IPI00413344.3 | Cofilin-2 | 90.96% | 6 | 22 | 4.79 | 25 | 1.81 | 13 | 6.93 | 11 | 166 |
| IPI00007068.1 | Actin-related protein 3B | 50.96% | 4 | 21 | 4.77 | 11 | | | | | 418 |
| IPI00012011.6 | Cofilin-1 | 93.98% | 15 | 29 | 4.67 | 56 | 1.75 | 37 | 5.34 | 56 | 166 |

*Note that SST14 is not included in this list because this protein only came to the fore as the most selective oAβ1-42-B interactor when the ≥6 amino acids and ≥3 peptides per protein requirements were waived.

cells exposed to Aβ (*Lovell et al., 2005*) (*Table 1*). Observations by others, which preceded this study, have provoked a hypothesis that a free N-terminus of Aβ1-42 might be critical for inducing neurofibrillary degeneration (*Jin et al., 2011*) and synapse loss (*Shankar et al., 2008*). We therefore explored how the orientation of Aβ-tethering might affect its protein-protein interactions. When the C-terminally biotinylated oAβ1-42-B bait was employed for affinity capture, a lower number of proteins was consistently observed to co-enrich with the bait and their ratios of co-enrichment (relative

to biotin-only negative controls) were generally lower than those observed with the N-terminally tagged B-oAβ1-42 bait (see also *Figure 1—source data 1*).

To determine the extent to which the binding of individual Aβ interactors was governed by Aβ aggregation, we next compared directly the interactome of proteins that bind to monomeric versus oligomeric preparations of the Aβ1-42-biotin bait. This experiment revealed a preference amongst the most highly enriched Aβ1-42 candidate interactors for binding to pre-aggregated bait peptides (*Table 1*). Exceptions represented the robust and preferred binding of histones H2, H3 and H4 to monomeric Aβ1-42 bait matrices (please see Supplementary Table 1 in *Supplementary file 1*). Moreover, a majority of proteins, which exhibited lower levels of Aβ1-42 co-enrichment in Experiments I and II, were observed to bind preferentially to monomeric Aβ bait peptides (*Figure 1—source data 1*).

## Pre-aggregated Aβ binds to SST14 but not to its preprosomatostatin precursor through a binding epitope in the N-terminal half of Aβ1-42

When we waived the requirement that peptides had to be at least six amino acids long to be considered for protein identification, a tandem mass spectrum assigned to the five amino acid sequence 'NFFWK' came to the fore, which exhibited pronounced preference for binding to C-terminally tethered oAβ1-42-B. A query of human genome databases revealed that this peptide, owing to its unusual composition, and despite its short length, could only have originated from the well-known paralogs preprosomatostatin or preprocortistatin (*Figure 2A*). This conclusion was strengthened by the fact that this peptide is naturally preceded by a tryptic cleavage site and the exquisite match between observed and in silico predicted fragment ions (*Figure 2B*). In light of the high intensity ion counts of fragments observed for this peptide and its robust co-enrichment with oAβ1-42-B in three biological replicates (*Figure 2C*), it first seemed puzzling that the identification of this protein group was not corroborated by other spectra that map to sequences outside of the 'NFFWK' sequence stretch. Searching for a plausible explanation, it became apparent that preprosomatostatin and preprocortistatin give rise to cyclic neuropeptide hormones through a series of posttranslational trimming steps, and the 'NFFWK' peptide is the only tryptic peptide derived from the mature hormone that is of sufficient length and distinct sequence to be readily identifiable by MS (*Figure 2A*). To further explore if binding had occurred to the mature neuropeptide, as opposed to the precursor, and to characterize the Aβ binding requirements, streptavidin-based affinity matrices were next side-by-side pre-saturated with biotin, N- or C-terminally biotinylated Aβ1-42 or an N-terminally truncated Aβ17-42-biotin bait. The subsequent application of an identical interactome analysis workflow as outlined above (but replacing TMT with iTRAQ isobaric labeling) (*Figure 1A*) corroborated the propensity of 'NFFWK' to bind to oAβ1-42-B but also established that the free N-terminus is indeed essential for this peptide-peptide interaction (*Figure 2D*). Although neither preprosomatostatin nor preprocortistatin had passed stringent filtering criteria required for inclusion in the Aβ interactome data tables (because their identification could not be based on at least three unique peptides with a minimum length of six amino acids), close inspection of Experiment III data under omission of these filters suggested that mature somatostatin was not only present in the dataset but represented the protein, whose levels were most selectively enriched in oAβ1-42-B affinity capture eluates. This conclusion was further corroborated by two additional PSMs that mapped to regions outside the SST14 neuropeptide sequence and one additional five amino acid peptide of the sequence 'TFTSC' that could be assigned (albeit not unambiguously) to the mature SST14 neuropeptide sequence itself. More specifically, consistent with the notion that Aβ1-42 interacts separately with the SST14 neuropeptide and preprosomatostatin precursor, the distributions of TMT signature ions derived from tryptic 'NFFWK' and 'TFTSC' peptides derived from the mature SST14 neuropeptide exhibited identical TMT signature ion profiles patterns that differed fundamentally from the respective TMT profiles of preprosomatostatin peptides which mapped to sequences upstream of the SST14 sequence domain (*Figure 2E*). In agreement with the interpretation that the 'NFFWK' had probably originated from SST14, not CST17, no peptides were observed in this or other experiments that could be uniquely assigned to preprocortistatin. Finally, these experiments repeatedly established that the 'NFFWK' peptide binds preferentially to oligomeric (pre-aggregated) but not monomeric Aβ1-42-biotin (*Figure 2E*).

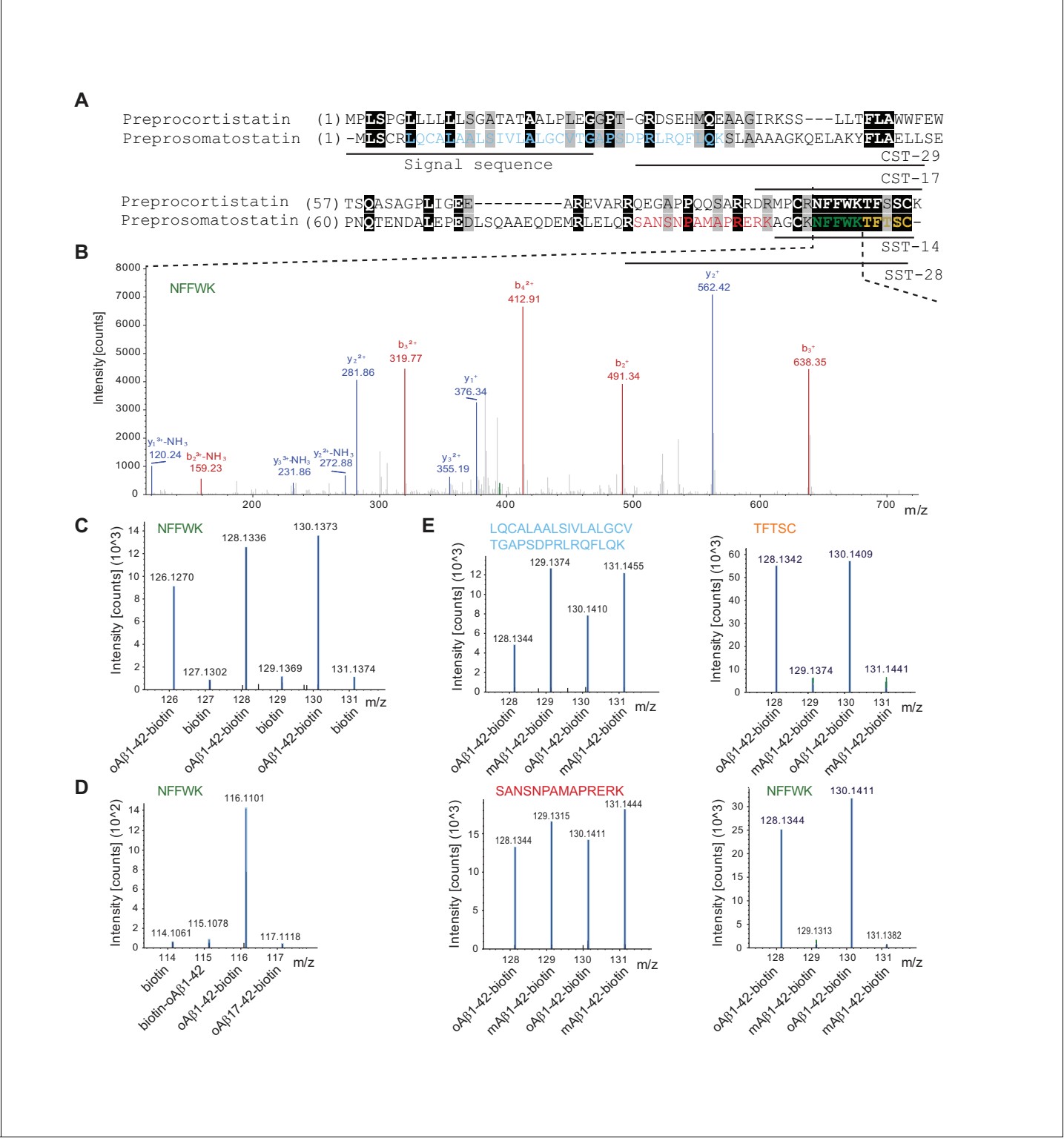

**Figure 2.** Discovery of somatostatin as a candidate interactor of oligomeric Aβ1-42. (**A**) Sequence alignment of preprocortistatin and preprosomatostatin. The signal sequence and the boundaries of the bioactive cortistatin and somatostatin peptides are indicated by horizontal bars. Identical residues are highlighted by black background shading, and peptide sequences observed by mass spectrometry are shown in colored fonts. (**B**) Example tandem MS spectrum supporting the identification of a peptide with the amino acid sequence 'NFFWK'. Fragment masses attributed to B- and Y- ion series are shown in red and blue colors, respectively. (**C**) Expanded view of MS3 spectrum derived from 'NFFWK' parent spectrum in interactome study based on oAβ1-42-biotin bait and biotin only negative control (Experiment II). In this view, the relative intensities of signature ions

*Figure 2 continued on next page*

*Figure 2 continued*

reflect the relative abundances of the 'NFFWK' peptide in the six side-by-side generated affinity purification eluate fractions. (D) SST/CST in human frontal lobe extracts binds to oAβ1-42-biotin but not to N-terminal biotinylated or truncated Aβ baits. iTRAQ signature ion intensity distribution in experiment probing the relative ability of four different biotin baits to capture SST/CST from human brain extract. The exclusive presence of a high intensity 116 ion indicates that the 'NFFWK' fragment spectrum, which gave rise to this peak distribution, was dependent on SST/CST exclusively associating with oAβ1-42-biotin. (E) Preferential binding of SST to pre-aggregated oAβ1-42. TMT signature ion intensity distributions of four MS3 spectra assigned to preprosomatostatin based on oligomeric or monomeric Aβ1-42-biotin baits (Experiment III). PSMs derived from SST-14 ('TFTSC' and 'NFFWK' peptides) had in common signature ion intensity distributions characterized by high intensity even-numbered TMT fragments. In contrast, signature ion intensity distributions of preprosomatostatin-derived tryptic peptides outside of the SST-14 coding region were relatively evenly distributed.

## Validation of SST14 binding to oAβ1-42

To further characterize the binding of SST14 to oAβ1-42, we next undertook a series of biochemical binding experiments. In a first reductionist approach, we explored if binding of SST14 requires co-factors or occurs directly and, therefore, can be observed with synthetic peptides. For this experiment, oAβ1-42 preparations (generated by incubating mAβ1-42 purified by gel filtration at 37°C for 2 hr with shaking at the speed of 700 rpm) gave rise to a pronounced signal of high molecular mass (HMM), not seen with mAβ1-42, when analyzed by denaturing SDS-PAGE and immunoblotting with the 6E10 antibody directed against an N-terminal Aβ1-42 epitope (*Figure 3A*). Next, streptavidin agarose matrices were pre-saturated with N-terminally biotinylated SST14 and exposed to unlabeled monomeric or oligomeric Aβ1-42 preparations. Finally, affinity capture matrices were boiled in Laemmli sample buffer and again immunoblotted with the 6E10 antibody (*Figure 3B*). The analysis revealed in samples which had been exposed to oAβ1-42 robust 6E10 signals both in the HMM and low mass region of the blot, the latter migrating at the same level as mAβ1-42. Because very little low mass signal was observed in samples, which had been incubated with mAβ1-42, we concluded that the low mass signal in oAβ1-42 exposed samples most likely reflected a release of a fraction of Aβ1-42 molecules from SST14-bound oAβ1-42 aggregates under the harsh denaturing analysis conditions. Taken together, this experiment validated that SST14 binding only occurs when Aβ1-42 is available in oligomeric form and established a direct mode of interaction.

To further validate the SST14-oAβ1-42 interaction and derive a binding constant we employed fluorescence resonance energy transfer (FRET) methodology based on a pairing of an SST14 donor and Aβ1-42 acceptor labeled with Edans and TMR fluorophores, respectively. To enable the formation of Aβ1-42 oligomers and aggregates, fluorescence spectra were recorded following a 24 hr incubation. Following excitation at a wavelength of 335 nm, the recording of independent fluorescence spectra of donor or acceptor peptides over a window of wavelengths spanning 350 nm to 650 nm gave rise to the expected Edans and TMR fluorescence intensity maxima at 493 nm and 596 nm, respectively (*Figure 3C*). Mixing of the two peptides caused profound quenching of the donor signal, concomitant with an increase in acceptor fluorescence, indicating energy transfer. The acceptor sensitization at 596 nm discouraged an interpretation whereby the donor quenching merely reflected an unspecific amorphous co-precipitation of SST14 with Aβ. Next, this experimental setup was exploited to derive a first indication of the binding constant that governs the interaction between SST14 donor and Aβ1-42 acceptor. To this end, various concentrations of unlabeled SST14 were added to the reaction mixture in a FRET competition experiment to determine the concentration (EC$_{50}$) for restoring donor fluorescence (*Figure 3D*). A half-maximal effective concentration of 12.7 μM SST14 could be deduced from the recorded graph. In contrast, addition of [Arg8]-vasopressin (AVP), a cyclic neuropeptide of similar size and biochemical characteristics as SST14, did not restore donor quenching in a control experiment, further strengthening the conclusion that the energy transfer reflected a specific interaction between the SST14 donor and Aβ1-42 acceptor.

## SST14 and CST17 delay Aβ1-42 aggregation in a Thioflavin T fluorescence assay

To explore the influence of SST14 on Aβ1-42 more rigorously, a Thioflavin T (ThT) fluorescence assay was applied that incorporated previously reported methodology advancements (*Hellstrand et al., 2010*), including the removal of residual Aβ1-42 aggregates by size-exclusion chromatography

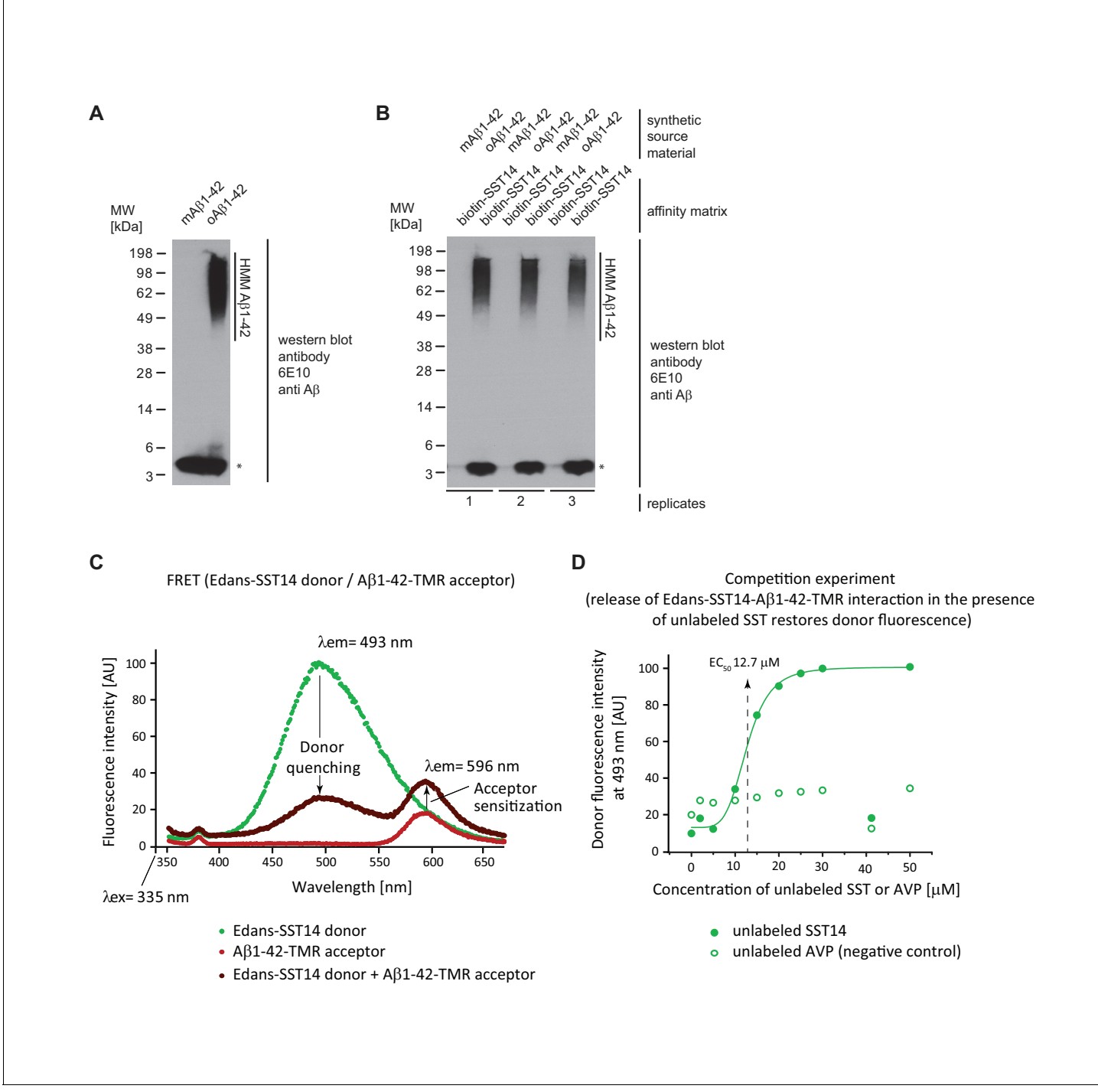

**Figure 3.** Validation of SST binding to oAβ1-42 but not to mAβ1-42. (**A**) Western blot analysis of synthetic mAβ1-42 and oAβ1-42. The asterisk designates a signal caused by the partial release of mAβ1-42 from high molecular mass (HMM) oAβ1-42 in the presence of SDS. (**B**) Biotin-SST affinity capture of oAβ1-42 but not mAβ1-42. (**C**) Evidence for fluorescence energy transfer between (FRET) between Edans-SST14 donor and Aβ1-42-TMR acceptor. Samples containing donor and acceptor peptides at 20 µM concentrations were incubated overnight at physiological pH. Note the profound quenching of the donor signal and increase in acceptor fluorescence relative to negative control preparations that contained only donor or acceptor peptides but were otherwise treated identically. (**D**) Competition FRET analysis based on configuration shown in panel 'D' but with unlabeled SST14 or the negative control AVP peptide being added at varying concentrations to the assay mix. Note the rescue of donor fluorescence in the presence of unlabeled SST14 but not AVP.

immediately prior to the recording of ThT fluorescence spectra (*Figure 4A*). The application of this method revealed that the presence of 15 µM SST14 (sourced from two independent vendors) in the reaction mix reproducibly extended the lag phase of Aβ1-42 aggregation by more than two hours and lowered the signal amplitude during the subsequent stationary phase (*Figure 4B*). The presence of an intact disulfide bridge within SST14 was not required for this delay to occur. No delay was observed when SST14 was replaced by the AVP negative control peptide. In further variations to this experimental setup it was established that pre-incubation of SST14, at a concentration which causes this peptide to pre-aggregate (see also *Figure 4—figure supplement 1*), did not affect its ability to delay ThT incorporation into Aβ1-42 aggregates (*Figure 4C*). Importantly, the SST14-dependent lag phase extension correlated directly with the concentration of SST14 in the reaction mix (*Figure 4D*). Moreover, a similar lag phase extension and/or reduction in ThT fluorescence intensity was not observed when the Aβ1-42 peptide was replaced in the same assay with amylin, another well-known amyloidogenic peptide of similar size (37 amino acids). More specifically, the presence of SST had no influence on the ThT fluorescence curve collected with a reaction mix containing 10 µM amylin. This result was observed even when SST was added to the reaction well at a considerably higher concentration of 100 µM (*Figure 4—figure supplement 2*). Finally, an experiment in which SST14 was replaced by CST17 revealed that the latter peptide is even more potent in its ability to interfere with the Aβ1-42 aggregation-dependent ThT incorporation. Thus, whereas the addition of 15 µM SST14 delayed Aβ1-42 aggregation by a few hours, addition of CST at the same concentration completely abrogated ThT incorporation (*Figure 4E*).

At concentrations above 3% w/w (~20 mM) the SST14 peptide had been observed to acquire amyloid characteristics, including Congo Red birefringence (*van Grondelle et al., 2007*). In our hands, even considerably lower concentrations of SST14 exceeding 250 µM reproducibly caused SST14 to incorporate ThT following a 2 hr lag phase (*Figure 4—figure supplement 1*). We used this property to determine if the presence of monomeric or oligomeric 1 µM Aβ1-42 influences SST aggregation. Note that the presence of 1 µM Aβ1-42 in this experiment does not contribute to ThT incorporation in this reversed assay configuration. More specifically, whereas the presence of 15 µM merely causes a profound lag phase extension (*Figure 4D*) the addition of 250 µM SST14 abolishes the ability of 1 µM Aβ1-42 to form aggregates that can incorporate ThT (not shown). Also note that in contrast to the first assay configuration, when low micromolar concentrations of SST14 were added to 1 µM Aβ1-42, in this second configuration of the ThT assay, substoichiometric quantities of 1 µM Aβ1-42 were added to a large excess of 250 µM SST. Strikingly, this analysis revealed that the presence of 1 µM Aβ1-42 caused a profound delay in SST14 aggregation-dependent ThT incorporation but only when Aβ1-42 had been allowed to pre-aggregate (*Figure 4—figure supplement 1B*). Taken together, these experiments corroborated the notion that the interaction of SST14 with Aβ1-42 relies on the presence of pre-aggregated oAβ1-42.

## Binding of SST14 or CST17 precludes detection of Aβ1-42 with commonly used antibodies

To determine if the presence of SST14 or CST17 simply prevented Aβ1-42 aggregation, or had caused the formation of alternative heteromeric complexes that failed to incorporate ThT, samples prepared as described in the previous section (but under omission of ThT) were analyzed by Western blotting using antibodies targeting N-terminal, central or C-terminal epitopes within Aβ1-42 (*Figure 5A*). Initially, we were interested in observing the influence of incubation time on sample composition (*Figure 5B*). As expected, the incubation of monomeric Aβ1-42 alone over a span of 18 hr caused its signals to gradually shift to higher molecular mass bands, reflecting the appearance of Aβ1-42 aggregates. In the presence of SST14 or CST17, 6E10-reactive signals that migrated at the level of monomeric Aβ1-42 also disappeared over the course of 18 hr albeit at a slower rate than observed with Aβ1-42 alone, most likely reflecting the delayed aggregation (*Figure 4D and E*). Strikingly, however, in the presence of SST14 or CST17 only very weak HMM signals of lower apparent molecular weight (50-70 kDa) than those seen in preparations of Aβ1-42 alone (55-200 kDa) could be seen following 18 hr incubation (*Figure 5B*, lanes 8 and 10). To further explore the apparent disappearance of 6E10-reactive bands during this time-course experiment, we next repeated the 18 hr incubation of Aβ1-42 in the presence of varying concentrations of SST14 or CST17. This experiment documented again an inverse correlation of the signal intensity of SST14 and CST17 concentrations and 6E10-reactive blot signals (*Figure 5C*). This effect was not limited to detection by the 6E10

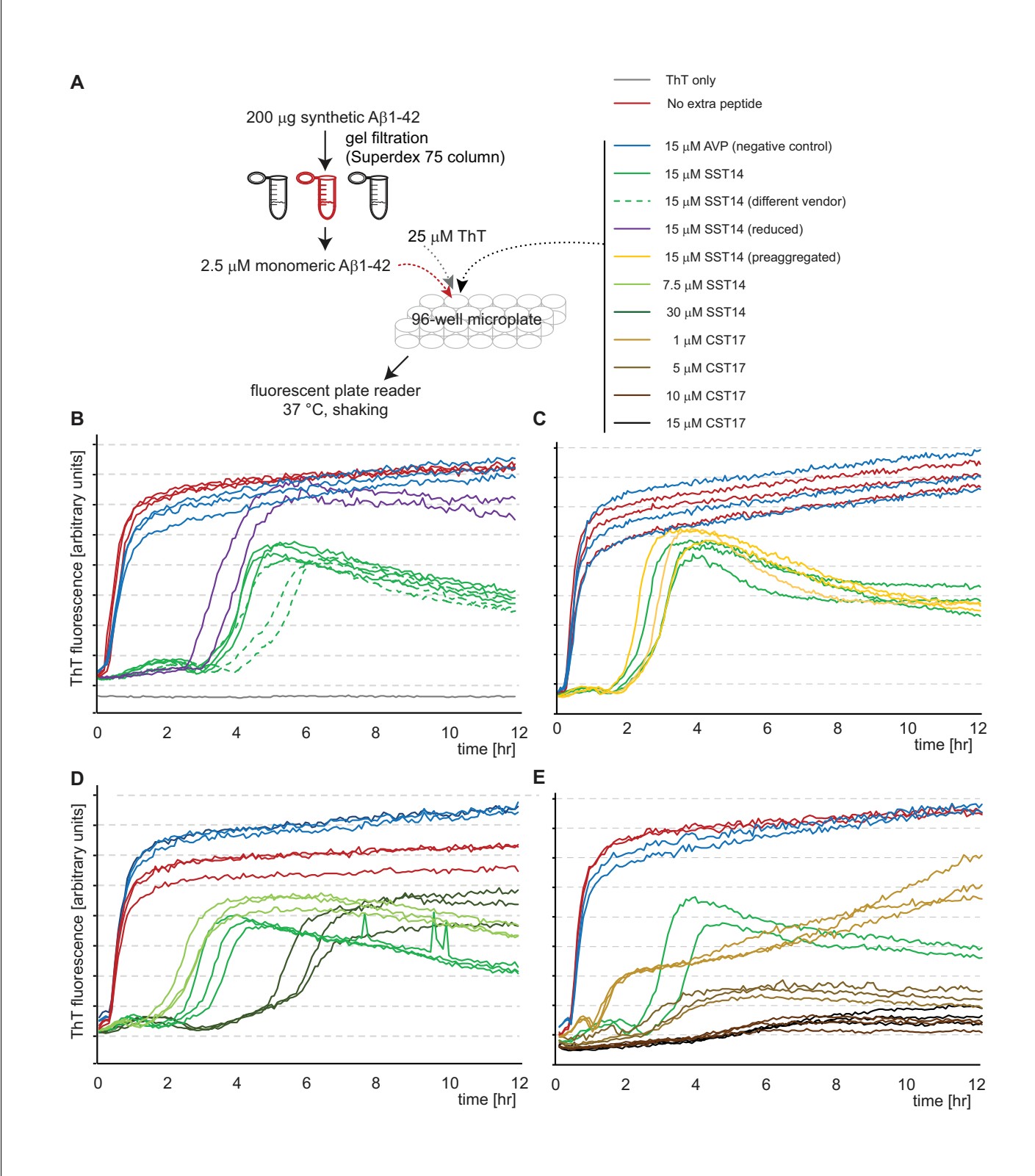

**Figure 4.** SST14 and CST17 delay Aβ1-42 aggregation in ThT fluorescence assay. (**A**) Workflow of ThT-based aggregation assay. (**B**) Representative ThT fluorescence charts using synthetic Aβ1-42 alone or in combination with SST14 or negative control AVP peptides. Note that SST14 alone does not contribute to ThT fluorescence in this assay at these relatively low concentrations. (**C**) Pre-aggregation of SST14 did not alter its effect on Aβ1-42 aggregation. (**D**) Evidence that the SST14-dependent delay in Aβ1-42 aggregation is SST14 concentration dependent. (**E**) Like SST, CST17 causes a

*Figure 4 continued on next page*

Figure 4 continued

concentration-dependent inhibition of Aβ1-42-dependent ThT fluorescence. Note that at 15 μM concentrations, CST17 appears more potent in this regard than SST14. Please see legend for experimental conditions.

The following figure supplements are available for figure 4:

**Figure supplement 1.** Oligomeric, but not monomeric, Aβ1-42 interferes with SST14-dependent ThT incorporation.

**Figure supplement 2.** The presence of SST14 does not affect the ThT fluorescence curve of amylin.

antibody but was similarly observed with antibodies targeting a central Aβ1-42 epitope (4G8) or the very C-terminus of Aβ1-42 (12F4) (not shown). The disappearance of signals could not be attributed to some unexpected disappearance of the Aβ1-42 peptide in these samples, because silver-stain analyses of the same fractions revealed relatively stronger Aβ1-42 signals in CST17 and SST14 containing samples. Rather, all available data suggest that the interaction of SST14 or CST17 masked the ability of these antibodies to bind to Aβ1-42.

## Aβ1-42 forms a distinct 50-60 kDa SDS-stable complex in the presence of SST14

We next sought to compare the effect of SST14 (or CST17) on the aggregation of Aβ1-42 and Aβ1-40. To learn more about how the presence of SST14 (or CST17) affects Aβ aggregation, ThT assay fractions were in this experiment collected and further analyzed by Western blotting. To compensate for the abovementioned epitope masking effect of SST14 or CST17 (*Figure 5C*), these experiments were undertaken with 5 to 10 times higher peptide concentrations than those described above (*Figures 4* and *5*). As expected, at these higher concentrations, Aβ1-42 and Aβ1-40 exhibited shorter lag phases in the ThT fluorescence assay of 30 min and 45 min, respectively (*Figure 6A*). Consistent with the ThT fluorescence data collected at lower peptide concentrations (*Figure 4*), the presence of CST17 (or SST14, not shown), but not AVP, extended the lag phase of Aβ aggregation considerably (by approximately one hour). However, except for slightly longer lag phases observed in fractions containing Aβ1-40, there was no difference in Aβ1-42 and Aβ1-40 kinetics revealed by this assay. Next, Western blot analyses of ThT assay fractions were undertaken without or with prior boiling in SDS. Interestingly, Aβ1-42-containing samples subjected to the milder treatment, which had been incubated for 18 hr in the presence of SST14 or CST17, gave rise to 6E10-reactive bands that migrated at 50-60 kDa, not seen in samples incubated either without these peptides or with AVP (*Figure 6B*, lanes 2 and 3). These intermediate-sized 6E10-reactive bands were even more pronounced when the same samples had been boiled in the presence of SDS (*Figure 6B*, lanes 10 and 11) but escaped detection with an antibody directed against the C-terminus of Aβ1-42 (*Figure 6B*, lanes 6 and 7). Interestingly, similar bands were not observed when Aβ1-40 served as the substrate for aggregation (*Figure 6B*, lanes 22 and 23), suggesting that the two most C-terminal residues of Aβ1-42 confer properties that are essential for the formation of the 50-60 kDa oligomeric assemblies. Only when SDS was present in the sample buffer, low mass bands appeared in the Western blot that migrated at a level expected for the monomeric Aβ peptide, consistent with the interpretation that these peptides were released from larger oligomeric or fibrillar aggregates under denaturing conditions (see also *Figure 5*). Strikingly, in samples that contained SST14 (or CST17) a fraction of these low mass bands, less than what would be needed for their detection by silver-staining, were occasionally (in well-resolved gels) observed to migrate at levels expected for heterodimers of Aβ1-42 and SST14 (or CST17) (*Figure 6B*, lanes 5, 6, 10 and 11). Consistent with the 3-amino acid smaller size of SST14, relative to CST17, the respective heterodimers containing SST14 migrated slightly faster than those containing CST17. A five-fold increase in Aβ1-42 levels in the reaction mix further emphasized the appearance of these low mass heterodimers but also revealed strong signals migrating at the expected size of Aβ1-42 dimers (*Figure 6B*, lanes 13 to 16). Notably, whereas evidence for heterotrimers composed of two Aβ1-42 molecules and one SST14 (or CST17) molecule was never obtained, we occasionally observed signals that migrated at molecular masses expected for SDS-stable complexes consisting of three Aβ1-42 peptides linked to SST14 (or CST17) (*Figure 6B*, lanes 14

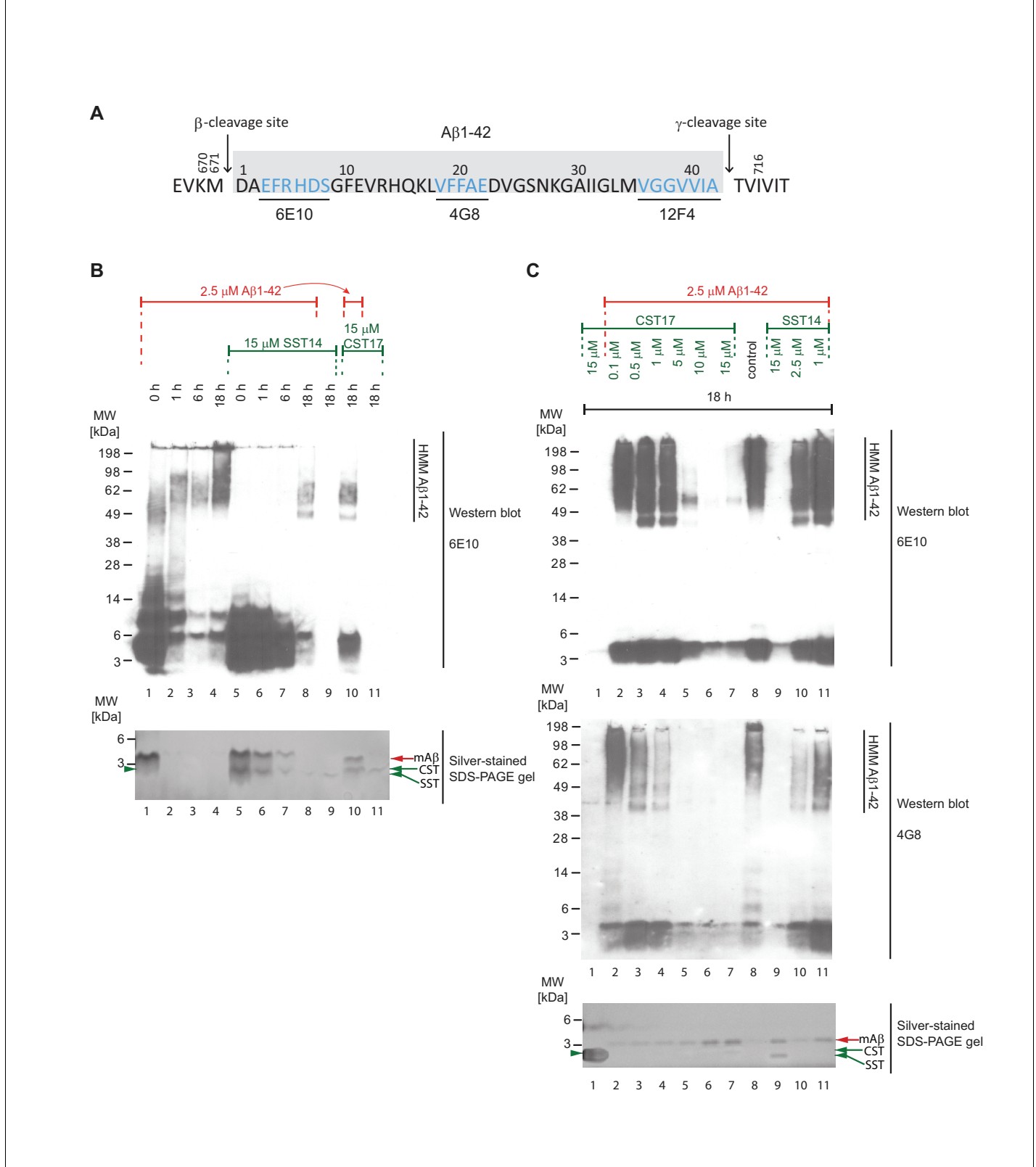

**Figure 5.** Binding of SST14 or CST17 precludes detection of Aβ1-42 with commonly used antibodies. (**A**) Schematic highlighting Aβ1-42 binding epitopes of antibodies used to generate this and the subsequent figure. (**B**) Western blot-based time-course analysis of Aβ1-42 aggregation in the presence or absence of SST14 or CST17. (**C**) SST14- and CST17-dependent masking of Aβ1-42 binding epitopes. Note the relatively more intense silver-stained bands of monomeric Aβ1-42 (red arrowhead) in samples containing the highest concentrations of SST14 or CST17 (lanes 6, 7 and 9), yet

*Figure 5 continued on next page*

*Figure 5 continued*

virtual absence of Aβ-specific immunoblot signals in the corresponding lanes. Note also the faster migrating band (green arrowhead) representing monomeric CST or SST observed in samples, which contained the highest concentrations of CST or SST (lanes 7 and 9).

The following figure supplement is available for figure 5:

**Figure supplement 1.** Full view of silver-stained gels depicted in *Figure 5*, panels B and C.

and 15). Corroborating this interpretation was the fact that bands matching this size were never observed in fractions lacking SST14 (or CST17) or containing AVP as a negative control peptide.

These biochemical analyses indicated the formation of SDS-stable oligomeric complexes of Aβ1-42 and SST14 (or CST17) made of building blocks comprising one or three Aβ1-42 peptides.

## Tryptophan-8 is critical for SST14-dependent perturbation of Aβ1-42 seeding and fibril growth

To delineate key residues and minimal components required for SST to interact with Aβ1-42, we next had a series of SST-derived peptides custom-synthesized that differed in one or two amino acids from the wild-type SST sequence or were truncated, thereby missing the internal disulfide bridge. Based on prior high-resolution NMR models available for the cyclic peptide and structure-activity relationship data derived from SST-receptor docking studies, we hypothesized that a striking hydrophobic 'belt' composed of three phenylalanines and a tryptophan, which is flanked by two lysine side-chains in the 3D rendering of SST (*Figure 7A,B and C*), might also be critical for binding to Aβ1-42. With regard to the juxtaposed phenylalanines, we were in particular interested to learn if the well-known ability of their side-chains to engage in pi stacking plays a role in the SST-Aβ1-42 interaction. Interestingly, the replacement of one or two of the phenylalanines to leucines did not rescue the lag phase extension phenotype in the ThT fluorescence assay, indicating that the aromatic nature of these residues is not essential for the interaction. However, when we replaced the single tryptophan present in position 8 both the lag phase extension and the reduction in total ThT fluorescence observed in the presence of wild-type SST were rescued (*Figure 7D*). This outcome was documented when tryptophan was replaced with alanine, proline, histidine or tyrosine. The fact that even the replacement with tyrosine rescued the lag phase extension phenotype indicates that tryptophan does not merely act in this context by providing an aromatic side-chain but suggests that other structural features present in tryptophan provide the necessary fit for binding to Aβ1-42 in this PBS-based system.

Analogous experiments with truncated SST peptides revealed that a peptide encompassing SST residues 5-11 can replace wild-type SST with regard to its effect on Aβ1-42 (*Figure 7E*), thereby upholding the conclusion that the ability of SST to form its internal disulfide bridge is not essential for this effect (*Figure 4B*). However, further stripping of one amino acid from each end of this peptide resulted in a SST6-10 peptide, which lacked the ability to influence Aβ1-42 aggregation.

We next compared Aβ aggregate size populations in the presence or absence of SST by nanoparticle tracking analysis (based on Nanosight technology [*Filipe et al., 2010*]). This orthogonal method documented that Aβ aggregates present in ThT aggregation assay fractions after 75 min of incubation were profoundly shifted toward smaller particle sizes when samples were co-incubated with SST (*Figure 7—figure supplement 1A and B*). Importantly, the replacement of SST with an SST-W8P negative control peptide did not result in this particle size shift toward smaller aggregates (*Figure 7—figure supplement 1C*), indicating that the altered particle size distribution is caused by the SST-Aβ interaction, rather than by some unspecific effect of the peptide.

Negative stain electron microscopy corroborated the reduction in aggregate size when Aβ1-42 was incubated with SST versus Aβ1-42 alone (*Figure 7—figure supplement 2*). Incubating 50 µM of Aβ1-42 under aggregation conditions resulted in the appearance of characteristic amyloid fibrils (*Figure 7—figure supplement 2A*), while in the presence of equimolar concentrations of SST only oligomeric structures were observed (*Figure 7—figure supplement 2B*).

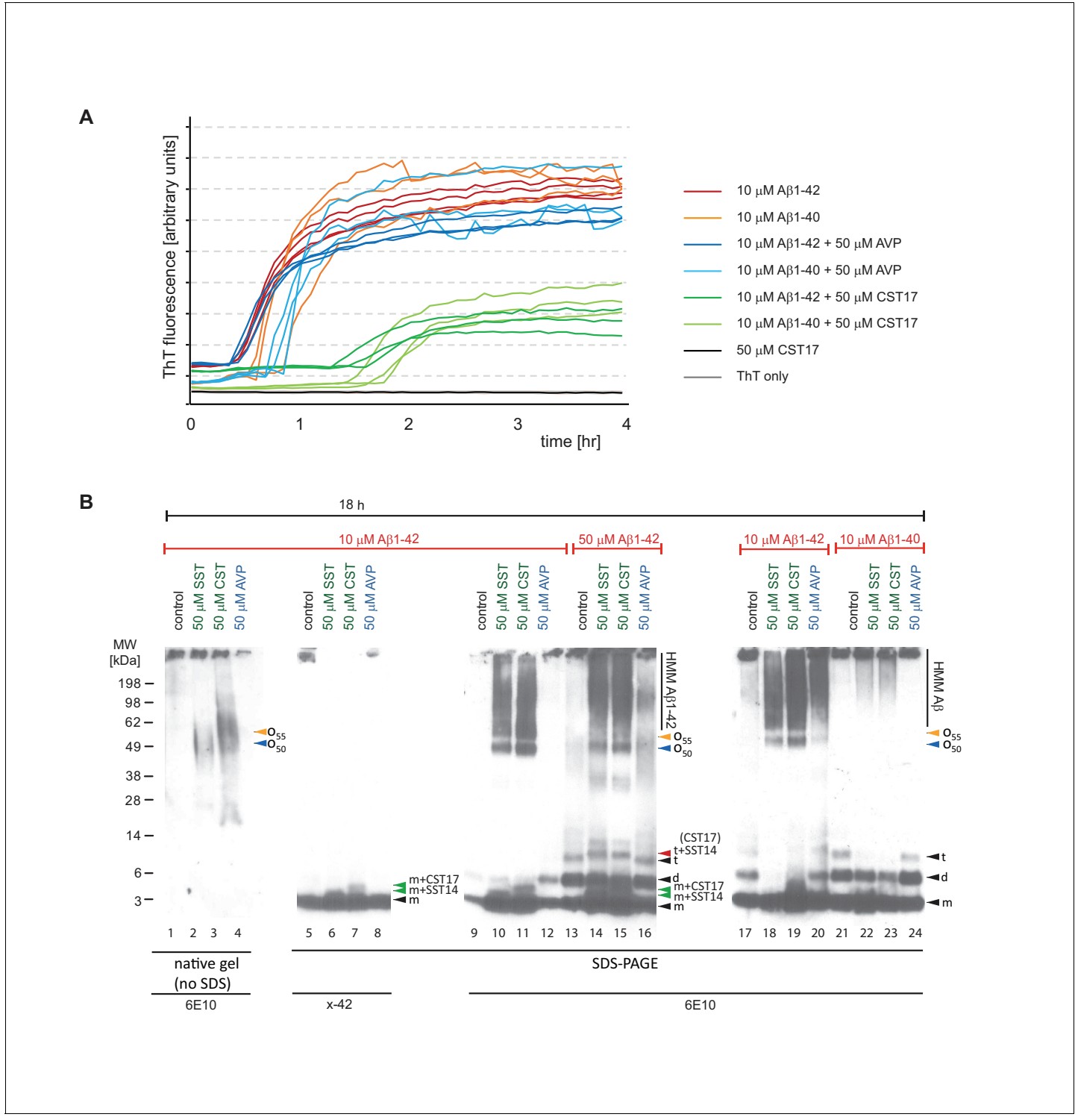

**Figure 6.** Aβ1-42 forms SDS-stable oligomeric complexes of 50-55 kDa in the presence of SST14 (or CST17). (**A**) In the presence of CST17 (or SST14, not shown) the ThT fluorescence curve or Aβ1-42 or Aβ1-40 is characterized by an extension of the lag phase and a reduction in ThT fluorescence. (**B**) CST17 (or SST14) co-assemble with Aβ1-42 into oligomers of 50-55 kDa that withstand boiling (lanes 2 and 3) but partially disintegrate in the presence of SDS. Immunoblot analyses with antibodies directed against the C-terminus (x-42) (lanes 6 and 7) or an N-terminal epitope (6E10) (lanes 10 and 11) revealed bands of 5-6 kDa, consistent with the existence of SDS-resistant heterodimeric complexes of mAβ1-42 and SST14 (or CST17). Note the well-defined oligomeric bands of 50 and 55 kDa (lanes 10 and 11) that were observed in samples derived from the co-incubation of SST14 (or CST17) with Aβ1-42 (lanes 10, 11, 18, 19). Note also that signals interpreted to represent trimeric Aβ1-42 (**t**) (lanes 13-16), but not dimeric Aβ1-42 (**d**), can be seen to migrate slower in the presence of SST14 (or CST17) (lanes 14 and 15). Finally, intensity levels of homodimeric Aβ1-42 bands are reduced in the presence

*Figure 6 continued on next page*

*Figure 6 continued*

of SST14 (or CST17) (compare lanes 17 and 20 with lanes 18 and 19). Black arrowhead labeled with 'm', 'd', and 't' designate bands interpreted to consist of monomeric, dimeric and trimeric Aβ1-42. Green and red arrowheads were used to label bands interpreted to represent SDS-stable heteromeric building blocks consisting of SST14 (or CST17) bound to monomeric and trimeric Aβ1-42, respectively.

Taken together, these results demonstrated exquisite specificity of the SST-Aβ1-42 interaction and corroborated the conclusion that Aβ assemblies in the presence of SST are smaller than those observed following Aβ-only aggregation.

## SST14 and CST17 can potentiate tau hyperphosphorylation when added to the cell culture medium together with monomeric Aβ1-42

A 2009 milestone report revealed that signaling downstream of SST positively modulates transcription of the neprilysin gene (*Saito et al., 2005*). Consistent with the known role of neprilysin as one of only a few proteases known to contribute to Aβ1-42 degradation, the authors reported an inverse relationship between SST and Aβ1-42 levels in mice deficient for the somatostatin gene. To begin to assess the physiological significance of the interaction between Aβ1-42 and SST14 observed in this study, an experimental paradigm was needed that is both amenable to manipulation and can be provoked to exhibit a phenotype considered relevant in the context of AD research. The primary hippocampal neuron assay, first described in detail by *Jin et al. (2011)*, fits this description (*Figure 8A*) and was favored over an analysis based on the technically challenging enrichment of the relatively small number of GABAergic interneurons known to release SST naturally. In agreement with prior data by others, robust tau hyperphosphorylation was observed when mouse hippocampal neurons, which had been cultured for 18 days, were exposed to oAβ1-42 but not when mAβ was added to the cell culture dish (*Figure 8B*). For the purposes pursued here, we restricted the tau phospho-epitope characterization to monitoring occupancy of pThr231 (AT180) and pSer202/pThr205 (AT8) phospho-acceptor sites, two of several known sites that undergo robust hyperphosphorylation also in AD brains, because we had noted that these phospho-epitopes were most responsive to oAβ exposure in the original report (*Jin et al., 2011*). We then modified the assay by the concomitant addition of synthetic SST14, other cyclic control peptides, or bombesin, which does not carry an internal disulfide bridge but was observed to share aggregation characteristics with SST (*Maji et al., 2009*). In a series of follow-up experiments, we added a pre-aggregation step for the peptide and/or Aβ1-42, varied the pH or duration under which this pre-aggregation was conducted, or changed the duration of cellular exposure to the peptide mixture as described in detail in the figure legend (*Figure 8A,C,D and E*). Because SST is naturally released from cells through the regulated secretory pathway when densely packed granules fuse with the plasma membrane (*Maji et al., 2009*), their content does not dissolve instantly when it encounters the extracellular milieu but dissolves with a time-scale of hours (*Anoop et al., 2014*). We therefore reasoned that low micromolar SST concentrations, including residual SST aggregates of varying size, might also be locally encountered by neurons in the brain in proximity to SST release sites.

When primary hippocampal neurons were exposed to SST14 alone, levels of tau phospho-occupancy at pSer202/pThr205 (AT8) were slightly reduced relative to vehicle-treated control cells. A similar reduction in phospho-occupancy was seen when SST14 was replaced with SST28 (*Figure 8C*, lanes 2-5), the longer physiologically relevant and bioactive version of this peptide, or the dominant preprocortistatin-derived regulatory peptides, CST17 or CST29. The addition of these peptides together with oAβ1-42 did not prevent the anticipated tau hyperphosphosphorylation-inducing effect of oAβ1-42 exposure (*Figure 8C*, lanes 8-11). Interestingly, the SST- dependent reduction in tau phosphorylation at the AT8 site, which was also observed when SST was subjected to a pre-aggregation step before its addition to primary hippocampal neurons (*Figure 8D*, lane 2), was not seen in the presence of other identically treated control neuropeptides, including AVP, bombesin and oxytocin (*Figure 8D*, lanes 3-5). Most remarkably, however, both SST14 and CST17, but not the aforementioned control peptides, were observed to potentiate tau hyperphosphorylation under the latter conditions when Aβ1-42 was concomitantly added in monomeric form to the cell culture medium (*Figure 8D*, lanes 7 and 8). This observation was notable on several grounds: (i) it indicated that SST14 and CST17 might be equivalent also in this experimental paradigm; and (ii) it revealed a

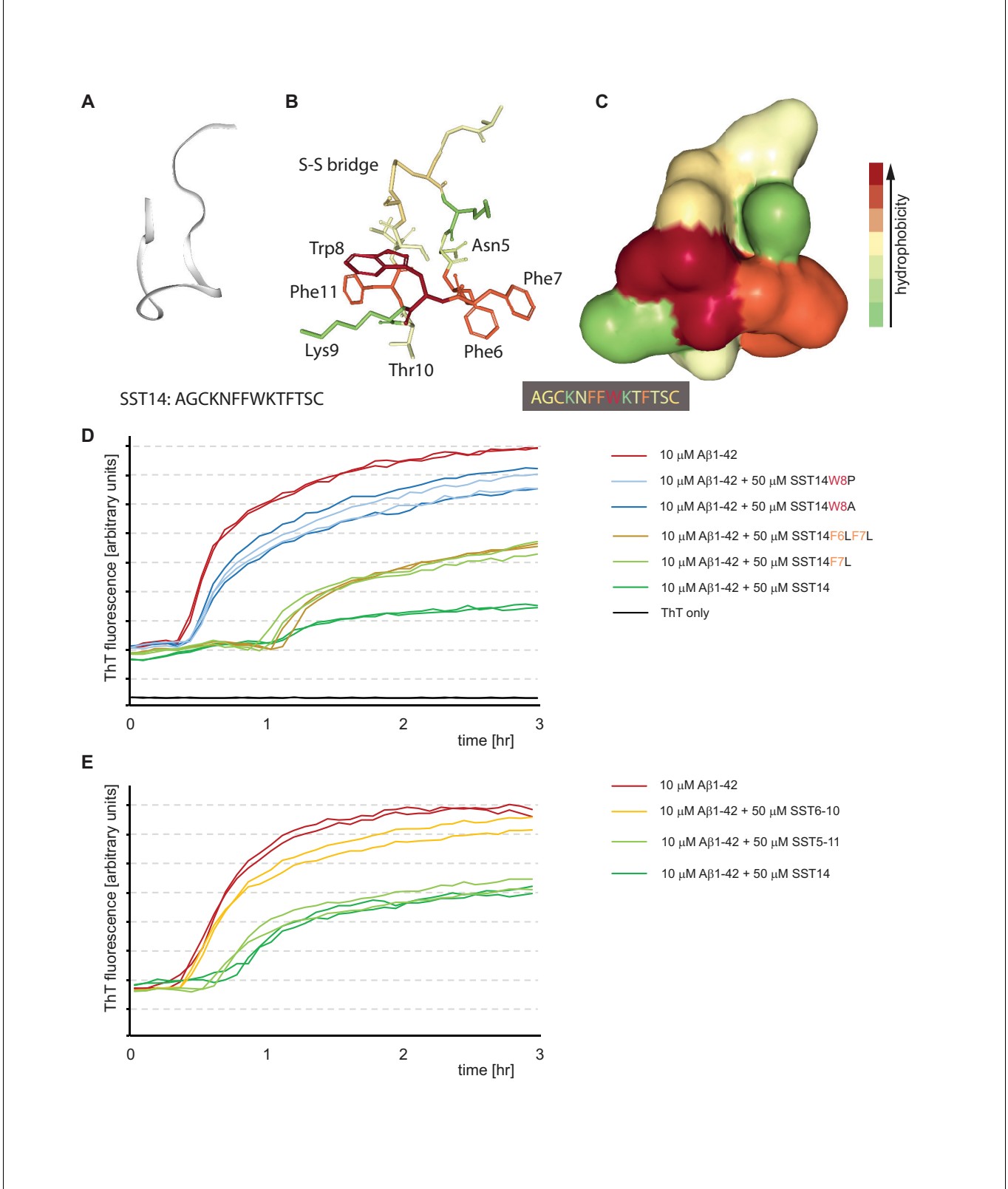

**Figure 7.** Tryptophan-8 in SST14 sequence is essential for lag phase extension of Aβ1-42 in ThT incorporation assay. (**A–C**) NMR-structure of SST14 in 5% D-mannitol (RCSF PDB structure ID: 2MI1, Model 1), adapted from *Anoop et al. (2014)*. Renderings were generated in NGL 3D viewer (powered by MMTF). (**A**) Backbone of SST14 emphasizing secondary structure. (**B**) and (**C**) Stick and surface models of SST14 with coloring emphasizing relative

*Figure 7 continued on next page*

*Figure 7 continued*

hydrophobicity. (D) and (E) Thioflavin T absorbance assay data based on SST point mutants and deletion constructs, respectively. Please see legend for sample compositions.

The following figure supplements are available for figure 7:

**Figure supplement 1.** In the presence of SST, Aß1-42 forms smaller quaternary assemblies.

**Figure supplement 2.** Negative stain electron microscopy of A$\beta$1-42 and A$\beta$1-42–SST14 complexes.

novel way in which tau hyperphosphorylation can be stimulated that, remarkably, was reliant on the co-addition of monomeric, not oligomeric, A$\beta$1-42 to the cell culture medium.

The objective of a series of subsequent experiments was to better define the experimental conditions that needed to be met for this SST-dependent tau hyperphosphorylation to occur. This line of investigation was pursued because, in contrast to the oA$\beta$1-42-induced tau hyperphosphorylation phenotype, which was robustly observed, SST-dependent tau hyperphosphorylation following co-administration of monomeric A$\beta$1-42 was not observed every time. Several plausible explanations can be invoked to explain such an experimental behavior, including reliance on experimental paradigms that are governed by threshold-effects of critical parameters or involve a stochastic component. Our current results argue against a stochastic limitation and are instead consistent with the view that once conducive conditions are met, the phenotype can be reliably observed. This view is, for instance, corroborated by one series of experiments, in which SST14 pre-aggregation was allowed to proceed for 24 or 48 hr. This experiment led to a tau hyperphosphorylation phenotype in three independent biological replicates only when the 48 hr pre-aggregated SST14 was added to the cell culture dish (*Figure 8E*, lanes 5, 8 and 11). As before, this molecular phenotype depended on the co-addition of monomeric A$\beta$1-42 and was again not observed when monomeric A$\beta$1-42 was added to the cell culture medium alone (*Figure 8E*, lanes 3, 6 and 9).

## Discussion

The current study was conducted with the intent to produce an in-depth inventory of proteins within the human brain that oA$\beta$1-42 binds to. The study made use of a hypothesis-free discovery approach that capitalized on advanced workflows for the high-pressure nanoflow reversed-phase separation and relative quantitation of peptides, as well as recent improvements to mass spectrometry instrumentation. Taken together, these advances afforded an unprecedented depth of analysis of oA$\beta$1-42 interactors in three independent interactome datasets, each undertaken with three biological replicates of samples and controls. More specifically, these analyses facilitated the direct comparisons of binders to oA$\beta$1-42-baits that were tethered to affinity capture matrices through N- or C-terminal biotin moieties, or had been captured in oligomeric versus monomeric form. From the large amount of data the study generated, several observations stand out: (1) despite its small size, oA$\beta$1-42 baits were observed to bind reproducibly to more than one hundred proteins in all analyses undertaken; (2) in particular, when oA$\beta$1-42 was conjugated to the affinity matrix through an N-terminal biotin moiety, a large number of candidate interactors co-enriched with the bait, corroborating a scenario whereby the hydrophobic C-terminal amino acid sequence stretch within A$\beta$1-42 is probably mostly protein- or lipid-bound; (3) the small cyclic peptide somatostatin, which has previously been implicated in the etiology of AD but had not been shown to interact with A$\beta$, was observed to bind directly to oA$\beta$1-42 in several orthogonal biochemical assays; (4) the presence of SST14 was observed to compromise antibody-based detection of A$\beta$1-42; (5) distinct assemblies of 50-60 kDa, reminiscent of the previously reported A$\beta$*56, were robustly generated in the presence of SST in reaction mixtures with A$\beta$1-42 but not with A$\beta$1-40; and (6) whereas A$\beta$ alone does not induce tau hyperphosphorylation in a primary hippocampal neuron assay, unless it is added to the cell culture medium in a pre-aggregated form, the concomitant exposure of cells to monomeric A$\beta$ and pre-aggregated SST can induce tau hyperphosphorylation at key AD-associated phosphorylation sites.

Several of the oA$\beta$1-42 binders observed in this work, including clusterin (*Narayan et al., 2011*; *Ghiso et al., 1993*), 17-$\beta$-hydroxysteroid dehydrogenase X (HSD10) (*Yan et al., 1997*), the ATP

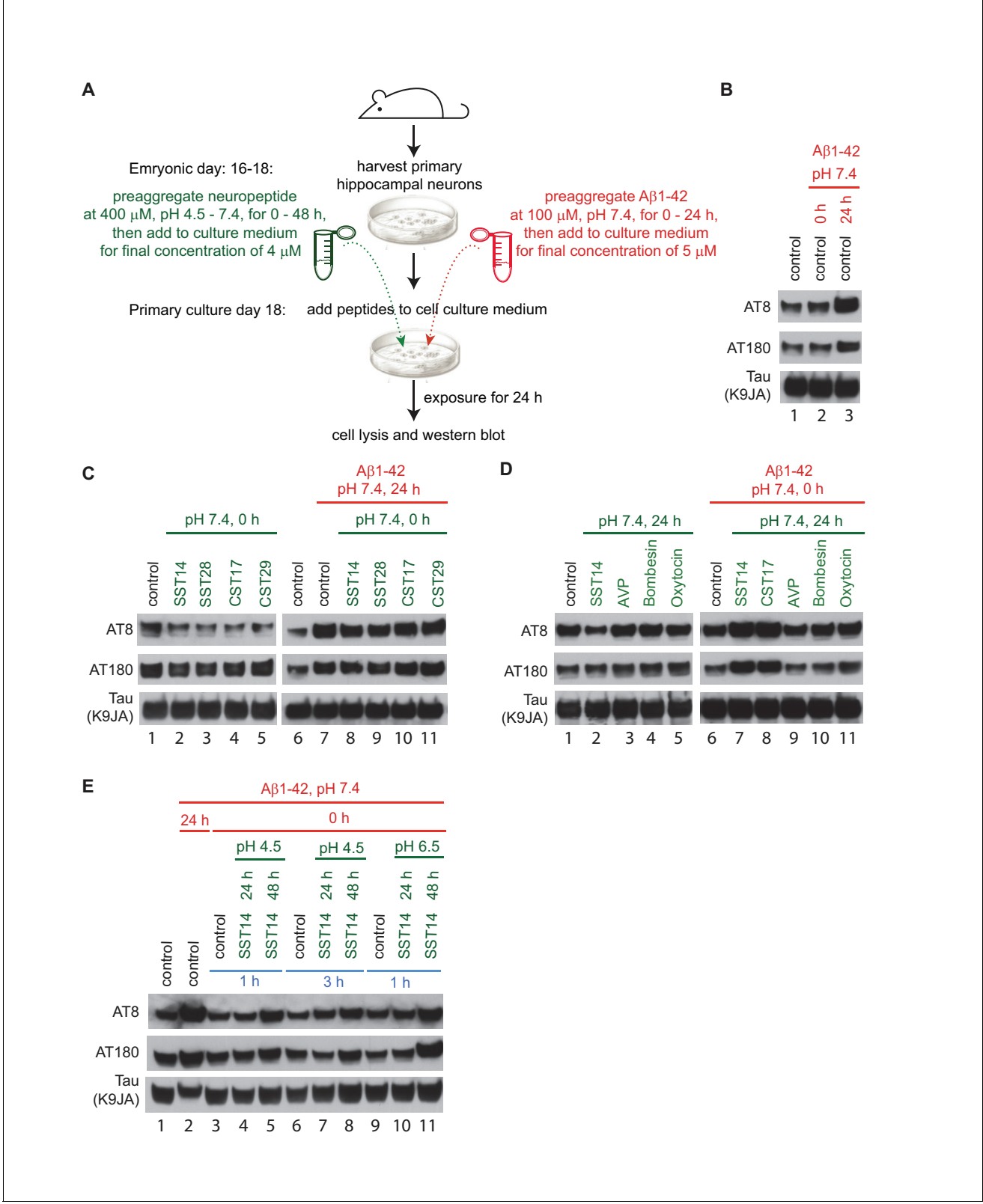

**Figure 8.** Exposure of primary hippocampal neurons to SST14 and Aβ1-42 can potentiate phosphorylation of tau at sites known to undergo hyperphosphorylation in AD. (**A**) Workflow of primary hippocampal neuron assay. Please see methods section for details on peptide preparation. (**B**) Addition of preaggregated (24 hr) but not monomeric (0 hr) Aβ1-42 to primary hippocampal neurons causes within 24 hr an increase in tau phospho-occupancy at AT8 and AT180 phosphorylation sites, in the absence of an effect on total tau levels (K9JA). (**C**) Addition of SST or CST bioactive peptides

*Figure 8 continued on next page*

*Figure 8 continued*

in the absence of Aβ1-42, leads to a reduction in tau phospho-occupancy. (**D**) Addition of SST14 or CST17, but not negative control cyclic peptides, can potentiate tau phospho-occupancy in the presence of monomeric Aβ1-42. (**E**) Tau hyperphosphorylation observed following addition of SST14 and mAβ1-42 to the cell culture appears to depend on conducive SST14 pre-incubation conditions. In all panels red and green font designates parameters (pH and pre-aggregation periods) applied to the preparation of Aβ1-42 and other neuropeptides, respectively. Peptides were added separately to the cell culture medium except for samples 3-11, Panel E, when peptides were mixed and pre-incubated for an additional 1 or 3 hr. Exposures of cells to peptides were for 24 hr.

synthase complex (*Schmidt et al., 2008*), GAPDH (*Verdier et al., 2005*), and the prion protein (*Laurén et al., 2009*), had previously been proposed to represent Aβ interactors. HSD10, also known as ERAB, 3-hydroxyacyl-CoA dehydrogenase type-2 or Aβ-binding alcohol dehydrogenase (ABAD), initially emerged from a yeast two-hybrid screen as an Aβ binding protein (*Yan et al., 1997*). Follow-up validation work, which preceded this study, generated evidence that this interaction may contribute to Aβ-mediated mitochondrial toxicity (*Lustbader et al., 2004*; *Yao et al., 2011*; *Valaasani et al., 2014*). The prion protein, which has been described as a major cell surface receptor for oligomeric Aβ (*Laurén et al., 2009*; *Chen et al., 2010*), was in this report found to co-enrich with oAβ1-42 tethered to the affinity matrix at its N-terminus but did not interact with C-terminally tethered oAβ1-42. A few candidate oAβ1-42 interactors revealed in this study, including the ubiquitin carboxy-terminal hydrolase L1 (Uch-L1) and the ras-related nuclear protein (RAN), had previously been proposed to play roles in the molecular etiology of AD but the mechanisms of their involvement have remained speculative. More specifically, both Uch-L1 and RAN were reported to be downregulated in AD (*Poon et al., 2013*; *Choi et al., 2004*; *Mastroeni et al., 2013*), and overexpression Uch-L1 was observed to restore synaptic function in hippocampal slices treated with oAβ (*Gong et al., 2006*). It will be interesting to explore the possible contribution of a physical interaction between oAβ1-42 and these proteins to these observations. Finally, this study revealed many new candidate interactors that had neither been known to interact with Aβ nor had otherwise been tied to the etiology of AD. The significance of these interactions is not known at this time. It is hoped that this inventory of interactors will expedite knowledge regarding the molecular underpinnings of the disease.

In extensive follow-up experiments, we validated the surprising interaction of Aβ1-42 with SST14, also known as somatotropin release-inhibiting factor (SRIF). A conclusion whether CST17 was also present in the initial oAβ1-42 interactome dataset could not be made because, in contrast to the situation for SST14, no peptide was observed that uniquely identified CST17. However, the possibility that binding of oAβ1-42 may extend to CST17 is supported by our observation that, although not yielding qualitatively different results in any of the binding assays, this peptide exerted even more pronounced effects on Aβ aggregation than SST14.

Not only did the oAβ-SST14 interaction stand out as a rare example of an interaction between two relatively small peptides but it also was remarkable in its dependence on a freely accessible N-terminus of Aβ1-42. Interest in binders to the N-terminus is heightened by the recognition that the C-terminus of Aβ1-42 is rarely accessible in vivo due to its hydrophobic nature. Although initially captured at pH 8.0, a widely-used pH for interactome analyses, owed to its wide-spread use in radio-immunoprecipitation assay (RIPA) buffer formulations, the oAβ-SST14 interaction was in subsequent validation experiments observed to influence Aβ-dependent aggregation characteristics in a buffer intended to mimic the physiological environment, and also withstood the presence of a lower pH during FRET analyses. To our knowledge, the smallest protein Aβ had previously been known to bind to is MT-RNR2, also known as humanin, a 21-24 amino acid peptide coded within the mitochondrial 16S ribosomal RNA gene (*Hashimoto et al., 2001*). Humanin has no resemblance to bioactive SST14 at the amino acid level, has been shown to acquire a three-turn α-helical fold, and interacts with an Aβ binding epitope encompassing residues 17-28 (*Maftei et al., 2012*), a segment of Aβ not sufficient for binding to SST (*Figure 2D*). Several previous reports have shown that synthetic macrocycles can assemble into oligomeric assemblies (*Levin and Nowick, 2007*) characterized by tetrameric assemblies in which β-sheet dimers pair through complementary dry interfaces (*Levin and Nowick, 2007*; *Liu et al., 2011*; *Pham et al., 2014*) and have the ability to serve as building blocks of larger fibril-like supramolecular assemblies (*Pham et al., 2014*). In the context of their

shared predominant release into the extracellular synaptic clefts in the brain, the robust influence of SST14 on Aβ aggregation observed in this work recommends SST14 as a natural Aβ binding partner that could contribute to alternative Aβ conformers and the existence of Aβ strains.

An intriguing data point in this work has been the observation that Aβ1-42, but not Aβ1-40, assembles into distinct complexes of 50-60 kDa in the presence of SST (or CST). Aβ complexes of similar size, most often referred to as Aβ*56, have been observed in transgenic APP mouse models (**Lesné et al., 2006**) and human AD brains (**Lesné et al., 2013**), and their levels have been proposed to correlate with cognitive impairment and toxicity. Consistent failures to generate Aβ*56 in vitro have led to the proposition that its formation may require an unknown co-factor (**Larson and Lesné, 2012**). While the relationship of Aβ*56 to SST-dependent oAβ complexes of similar mass remains unresolved, the ability to generate Aβ assemblies of this size in vitro should aid their structural characterization and facilitate the raising of antibodies with the intent to evaluate their presence in patient samples. Moreover, the ability of SST to prevent the fibrillization of Aβ1-42 in vitro may have implications for the generation of toxic Aβ1-42 oligomers in AD patients. We further documented that the presence of SST profoundly diminished the detection of Aβ by some of the most frequently used antibodies. More work is needed to understand the molecular basis for this finding and gauge the extent to which its existence may have undermined Aβ detection in studies that preceded this work.

SST, originally identified as a factor that inhibits the secretion of pituitary growth hormone (**Brazeau et al., 1973**), is not a neophyte in the field of AD; a PubMed query conducted with the search terms 'somatostatin' and 'Alzheimer' returns more than 300 entries, including 60 review articles (e.g., (**Epelbaum et al., 2009**; **Burgos-Ramos et al., 2008**). Several observations stand out amongst the connections of SST to AD and should be considered in future efforts to understand the possible significance of the direct interaction between SST and Aβ1-42: (1) Already in 1980, it was observed that SST levels are conspicuously reduced in AD-afflicted brains (**Davies et al., 1980**), a finding which has been repeatedly confirmed by others (**Beal et al., 1985**; **Gahete et al., 2010**; **Gabriel et al., 1993**). More recently, it was shown that SST gene transcripts are amongst a relatively small number of transcripts whose abundance levels decline persistently throughout adult life (**Lu et al., 2004**) ; (2) A 1985 study reported a striking co-localization of SST-filled neurons with sites of amyloid plaque formation in AD (**Morrison et al., 1985**); (3) a manuscript published back-to-back documented that the expression of SST characterized a subclass of neurons in the brain, which were most vulnerable to forming neurofibrillary tangles and undergoing destruction in AD (**Roberts et al., 1985**); (4) two studies undertaken with Finnish and Chinese patient cohorts reached the conclusion that an SST gene polymorphism may increase the risk for AD (**Xue et al., 2009**; **Vepsäläinen et al., 2007**) and, (5) independent investigations revealed that SST can regulate Aβ catabolism by modulating the expression of neprilysin (**Saito et al., 2005**) and insulin-degrading enzyme (**Tundo et al., 2012**), two critical players in the best-characterized Aβ clearance pathways. More specifically, Aβ levels were reported to be approximately 1.5-fold higher in SST-deficient mice (**Saito et al., 2005**), an increase that did not lead to spontaneous plaque formation, thereby leaving the possible involvement of SST in this process currently untested. Once AD models become available in which SST has been eliminated, it will be informative to assess how their Aβ-related pathobiology is altered. It will also be interesting to investigate the constraints of the SST-oAβ interaction in more detail. Our current data suggest that this interaction depends on the presence of tryptophan 8 within SST in the context of flanking residues 5-11 and as such strongly overlaps the epitope within SST known to mediate binding to its cell surface receptors (**Veber et al., 1979**). Our results further suggest that binding does not occur between monomers of both peptides but may require at least one of the two binding partners to be present in oligomeric (pre-aggregated) form. This limitation is likely to pose technical difficulties for its detailed study but also offers the tantalizing prospect that future insights into the binding interface can be used to design compounds, which may selectively bind to the respective oligomeric forms of Aβ. Such compounds might find application in disease diagnosis, might be used to prevent the SST-Aβ interaction from forming, or could possibly be derivatized in ways that target the respective conformers, and possibly amyloid plaques, for destruction.

# Materials and methods

## Peptides

Synthetic human Aβ1-42 (catalog number AS-24224), Aβ1-40 (catalog number AS-24236), SST14 (catalog number AS-24277), [Arg8]-Vasopressin (catalog number AS-24289), Bombesin (catalog number AS-20665), Oxytocin (catalog number AS-24275), SST28 (catalog number AS-22902) and amylin1-37 (catalog number AS-60804) were purchased from Anaspec, Inc. (Fremont, CA, USA). The alternative synthetic human SST14 peptide (catalog number H-1490) sourced from a different vendor (see *Figure 4B*), CST17 (catalog number H-5536) and CST29 (catalog number H-6458) were obtained from Bachem Americas, Inc. (Torrance, CA, USA). The biotinylated Aβ peptides, including biotin-Aβ1-42, Aβ1-42-biotin, Aβ17-42-biotin, biotin-SST14 and truncated or mutant SST peptides were synthesized by LifeTein LLC (Hillsborough, NJ, USA). The Edans-SST donor and Aβ1-42-TMR acceptor peptides were in-house synthesized as described previously (*Bateman and Chakrabartty, 2011*).

## Antibodies

Primary antibodies used in this study were the anti-Aβ1-16 antibody (6E10) (catalog number 803015), the anti-Aβ17-24 antibody (4G8) (catalog number 800701), and the anti-Aβ1-42 antibody (12F4) (catalog number 805501), which were all sourced from BioLegend (San Diego, CA, USA) (note that epitope ranges embedded in vendor names for these antibodies are imprecise, and validated epitopes for the above antibodies are depicted in *Figure 5A*). The polyclonal antibody recognizing total Tau (K9JA) (catalog number A0024) was a Dako product (Agilent Technologies Canada, Mississauga, ON, Canada). The monoclonal antibodies recognizing Tau phosphorylated at Ser202/ Thr205 (AT8) (catalog number MN1020) or at Thr231 (AT180) (catalog number MN1040) were from Thermo Fisher Scientific (Burlington, ON, Canada).

## Western blot and silver staining

All Western blot reagents were purchased from Thermo Fisher Scientific (Burlington, ON, Canada). For denaturing gels (*Figures 3*, *5* and *6*), peptide samples were mixed with Bolt LDS Sample Buffer (catalog number B0007) in the presence of 2.5% 2-mercaptoethanol and boiled at 70 °C for 10 min before loading. The samples were separated on Bolt 12% Bis-Tris Plus gels (catalog number NW00125BOX) in MES SDS Running Buffer (catalog number NP0002) at 100 to 120 V for 1.5 to 2 hr. For native gels (*Figure 6B*), the peptide samples were mixed with Novex Tris-Glycine Native Sample Buffer (catalog number LC2673) without boiling before loading. The samples were separated on SDS-free 4-20% Tris-Glycine Mini Gels (catalog number XP04202BOX) in Novex Tris-Glycine Native Running Buffer (catalog number LC2672) at 150 V for 1.5 hr. 0.3 to 0.5 μg of Aβ peptide were loaded to each lane for both denaturing and native gels. For the analysis of hippocampal neuron lysates, samples were prepared and separated in the same way as described for denaturing gels except that 4% to 12% Bis-Tris gradient gels were used and 20 to 30 μg of total protein were loaded to each lane. For all immunoblot analyses, peptides were transferred to polyvinylidene difluoride (PVDF) membranes at 50 V in Tris-Glycine buffer containing 10-20% methanol for 1.5 to 2 hr. Membranes were blocked for 2 hr in conventional tris-buffered saline and 0.1% Tween 20 (TBST) containing 5% fat-free milk and probed overnight with the respective primary antibodies. After at least three washes with TBST, membranes were incubated for 2 hr with 1: 2000 to 1: 5000 diluted anti-mouse or anti-rabbit horseradish peroxidase-conjugated secondary antibodies (Bio-Rad Laboratories, Inc., Hercules, CA, USA). The band signals were visualized using enhanced chemiluminescence reagents (catalog number 4500875; GE Health Care Canada, Inc., Mississauga, ON, Canada) and X-ray films. Silver staining of gels was done with reagents from Pierce Silver Stain Kit (catalog number 24612) (Thermo Fisher Scientific, Burlington, ON, Canada) following the protocol provided by the manufacturer.

## Preparation of monomeric versus pre-aggregated Aβ, SST and other neuropeptides

Aβ peptides were dissolved in 1,1,1,3,3,3-hexafluoro-2-propanol (HFIP) at a concentration of 1 mg peptide per mL for one hour at room temperature, then dried in a centrifugal evaporator and stored

at −80°C until use. Monomeric Aβ peptides were prepared by dissolving peptides in dimethyl sulfoxide (DMSO) at a concentration of 2 mM and further diluting them in phosphate buffered saline (PBS, pH 7.4) to 100 µM, followed by centrifugation (14,000 g, 20 min) to remove traces of insoluble aggregates. Oligomeric Aβ was created by incubating the monomeric preparation at 4°C for 24 hr. The Aβ oligomers were then purified by centrifugation (14,000 g, 20 min). The supernatant containing Aβ oligomers was further diluted as required and indicated in figure legends describing specific experiments. When high-purity monomeric Aβ peptides at concentrations below 10 µM were required for interactome and ThT assay analyses, the solubilized Aβ preparations were passed through a size-exclusion column (see below for details). Pre-aggregated SST was generated by incubating 1 mM SST in PBS at 37°C for 1 hr with shaking at the speed of 700 rpm. For SST and other neuropeptides prepared for the hippocampal neuron culture assay, the peptides were dissolved in PBS (at pH 4.5, 6.5 or 7.4) at 400 µM without incubation (*Figure 8C*) or were incubated at room temperature for 24 or 48 hr (*Figure 8D and E*). 50 µM Aβ1-42 and 40 µM SST/other neuropeptides were mixed (*Figure 8D*) and incubated for 1 or 3 hr (*Figure 8E*) before they were added to cell culture medium. The final concentration of Aβ1-42 and SST/other neuropeptides in the medium were 5 and 4 µM, respectively.

## Affinity capture of Aβ1-42 and its binding proteins in human frontal lobe extracts

The biotinylated Aβ oligomers or monomers were captured on Streptavidin UltraLink Resin beads (Thermo Fisher Scientific, Burlington, ON, Canada) by overnight incubation in PBS at 4°C and continuous agitation on a slow-moving turning wheel. Additional negative control samples were generated by saturation of Streptavidin UltraLink Resin (catalog number 53113, Thermo Fisher Scientific, Inc.) with biotin. Subsequently, the bait peptide- or biotin-saturated beads were washed with Lysis Buffer (0.15% digitonin, 150 mM NaCl, 100 mM Tris, pH 8.0). Human frontal lobe tissue samples from individuals (two males and two females) who had died in their early 70s of non-dementia causes served as the biological source material. These samples were adopted from a former Canadian Brain Tissue Bank at the Toronto Western Hospital and are held in −80°C freezers in the biobank of the Tanz Centre for Research in Neurodegenerative Diseases. 1 g pieces each of these brain tissue samples were combined and homogenized in Lysis Buffer supplemented with Complete protease inhibitor cocktail (Roche, Mississauga, ON, Canada). Following the removal of insoluble debris by centrifugation for 30 min at 14,000 g, the protein concentration was adjusted to 2 mg/mL before the brain homogenates were added to the pre-saturated affinity capture beads for overnight incubation at 4°C. Following the affinity capture step, the affinity capture beads (100 µL per biological replicate) were extensively washed in three consecutive wash steps with a total of 150 mL of Lysis Buffer. Subsequently, the beads were additionally washed with 50 mL of 20 mM Hepes, pH 7.0, and transferred to Pierce Spin columns (catalog number 69705, Thermo Fisher Scientific, Inc.) to remove primary amines stemming from the Tris buffer and to prepare the samples for elution. Captured proteins were finally eluted by rapid acidification mediated by a solution comprising 0.2% trifluoroacetic acid and 20% acetonitrile in deionized water (pH 1.9).

## Affinity capture of synthetic Aβ1-42 on biotin-SST-saturated capture beads

N-terminally biotinylated SST14 was captured on Streptavidin UltraLink Resin beads (Thermo Fisher Scientific, Burlington, ON, Canada) at a concentration of 50 µM by overnight incubation in PBS at 4°C with continuous agitation on a turning wheel. Next, the biotin-SST14 saturated streptavidin agarose beads were incubated at room temperature for 4 hr with monomeric or oligomeric Aβ1-42 preparations (2.5 µM), prepared as described in section 'Preparation of monomeric versus oligomeric Aβ or SST'. Before and after the incubation, the beads were extensively washed in Lysis Buffer (0.15% digitonin, 150 mM NaCl, 100 mM Tris, pH 8.0). Finally, bound peptides were eluted from the affinity capture beads by 15 min boiling in Laemmli sample buffer and analyzed by Western blotting using NuPAGE gels as described in section 'Western blot and silver staining'.

## Size-exclusion chromatography of Aβ1-42

200 µg of monomeric Aβ1-42 were generated as described in the section 'Preparation of monomeric versus oligomeric Aβ or SST/other neuropeptides'. To further remove residual traces of Aβ1-42 oligomers, the soluble centrifugation supernatant was subsequently subjected to size-exclusion chromatography on a Superdex 75 10/300 GL column (GE Healthcare Life Sciences, Mississauga, ON, Canada) in ThT assay buffer (20 mM sodium phosphate, pH 8.0, 200 µM EDTA and 0.02% NaN₃) at the flow rate of 0.5 mL/min. Late eluting fractions, containing Aβ1-42 monomers, were collected on ice and the Aβ1-42 concentration in these fractions was determined by Western blot analysis, using known amounts of synthetic Aβ1-42 as signal intensity calibrants.

## Sample preparation for interactome analyses

Affinity-capture eluates were essentially processed as described before (*Gunawardana et al., 2015*; *Mehrabian et al., 2014*; *Jeon and Schmitt-Ulms, 2012*). Briefly, sample tubes were moved to a centrifugal evaporator to remove the organic solvent. Additional acidity of the sample was removed following the addition of water and continuous evaporation. Subsequently, protein solutions were denatured by the addition of 9 M urea (to achieve a final concentration of 6 M urea) and 10 min incubation at room temperature. Next, the pH was raised by the addition of 100 mM HEPES, pH 8.0, and proteins were reduced for 30 min at 60°C in the presence of 5 mM tris (2-carboxyethyl) phosphine (TCEP), and alkylated for 1 hr at room temperature in the presence of 10 mM 4-vinylpyiridine (4-VP). To ensure that the residual urea concentration did not exceed 1.5 M, protein mixtures were diluted with 50 mM tetraethylammonium bromide (TEAB), pH 8.0, to a total volume of 100 µL. Samples were then digested with side-chain-modified porcine trypsin (Thermo Fisher Scientific, Burlington, ON, Canada) overnight at 37°C. The covalent modifications of primary amines with isobaric labels provided in the form of tandem mass tag (TMT) reagents (Thermo Fisher Scientific, Inc.) or isobaric tagging for relative and absolute quantitation (iTRAQ) reagents (Applied Biosystems, Foster City, CA, USA) followed instructions provided by the manufacturers. Equal amounts of the labeled digests were pooled into a master mixture and purified with C18 (catalog number A5700310) or SCX (catalog number A5700410) Bond Elut OMIX tips (Agilent Technologies, Inc., Mississauga, ON, Canada) using manufacturer instructions. Peptide mixtures were finally reconstituted in 0.1% formic acid and analyzed by tandem mass spectrometry analysis on a Tribrid Orbitrap Fusion instrument. Instrument parameters during the data acquisition were as described in detail before (*Gunawardana et al., 2015*).

## Post-acquisition data analyses

The post-acquisition data analyses of interactome data sets was conducted against the human international protein index (IPI) database (Version 3.87) which was queried with Mascot (Version 2.4; Matrix Science Ltd, London, UK) and Sequest HT search engines within Proteome Discoverer software (Version 1.4; Thermo Fisher Scientific, Burlington, ON, Canada). Spectra exceeding a stringent false discovery rate (FDR) target of ΔCn of 0.05 for input data and a FDR of 0.01 for the decoy database search were detected and removed by the Percolator algorithm (*Käll et al., 2007*) as described before (*Gunawardana et al., 2015*). PEAKS Studio software (Version 6.0; Bioinformatics Solutions Inc., Waterloo, ON, Canada) was used to assess the reproducibility of nano-HPLC separations. A maximum of two missed tryptic cleavages and naturally occurring variable phosphorylations of serines, theonines and tyrosines were considered. Other posttranslational modifications considered were carbamylations, oxidation of methionines and deamidation of glutamines or asparagines. Mass spectrometry data sets have been deposited to the ProteomeXchange Consortium (*Vizcaíno et al., 2014*) via the PRIDE partner repository (*Vizcaíno et al., 2013*) with the dataset identifier PXD004867 and have been made fully accessible.

## FRET measurements

All fluorescence spectra were acquired with a Photon Technology International fluorescence spectrophotometer (QuantaMaster, HORIBA Scientific, London, ON, Canada) using a 1 cm quartz cuvette with an excitation wavelength of 335 nm and a 2 nm slit width at room temperature. Solutions containing 20 µM concentrations of Edans-SST14, Aβ1-42-TMR or a mixture of both fluorescently labeled peptides at pH 8.5 were rapidly adjusted to pH 5.2 and the resulting reaction mixtures

incubated overnight. Fluorescence emission spectra were recorded in a continuous window spanning 350 nm to 650 nm.

## FRET competition assay

FRET competition assays were performed using the fluorescence spectrophotometer described above. A solution of 2 µM A$\beta$1-42-TMR was mixed with increasing concentrations of SST14 or AVP (0-50 µM) at pH 8.5 followed by dropping the pH to 5.2 and incubating the resulting solutions overnight. To compete for binding to A$\beta$1-42-TMR, a solution of 2 µM Edans-SST was then added to each of the solutions at a final volume of 150 µL at continuous incubation of peptide mixtures at pH 5.2 overnight. Fluorescence spectra were recorded and binding curves were plotted against the concentration of unlabeled peptides. The curves were fitted with assistance of OriginPro 8.5 software (OriginLab Corporation, Northampton, MA) using nonlinear least squares fitting given by the equation:

$$y = (y_{max} - y_{min}/(1 + (x/EC_{50})n) + y_{min}$$

with $y$ = observed donor fluorescence, $x$ = concentration, and $n$ = Hill coefficient.

## Kinetic aggregation assay

Assay procedures were largely based on a protocol described by Sarah Linse's group (*Hellstrand et al., 2010*). Briefly, A$\beta$1-42, A$\beta$1-40 and/or SST14 were prepared in assay buffer (20 mM sodium phosphate, pH 8.0, 200 µM EDTA and 0.02% NaN$_3$) or in PBS, pH 7.4, at concentrations specified in the individual figures. Human amylin1-37 was initially dissolved in deionized water but then assayed under identical conditions as A$\beta$1-42. 100 µL of peptide solutions were supplemented with 25 µM thioflavin T (ThT) (catalog number T3516, Sigma-Aldrich Canada, Oakville, ON, Canada) and loaded into 96-Well Half-Area Microplates (catalog number 675096, Greiner Bio One International, Kremsmünster, Austria). The subsequent plate incubation proceeded at 37°C with shaking at 700 rpm for 4 of every 5 min in a microplate reader (CLARIOstar, BMG Labtech, Guelph, ON, Canada) for overall durations specified in individual figures. ThT fluorescence was measured every 5 min at excitation and emission wavelengths of 444 nm and 485 nm, respectively.

## Particle sizing

Particle size was analyzed using a NanoSight NS300 (Malvern Instruments, Southborough, MA, USA). Samples were vortexed for 40 s before being loaded into a luer lock syringe and infused at a constant rate through the sample chamber using an attached syringe pump (Malvern). Particle movement was captured for 60 s x 5 acquisitions at a rate of 25 frames per second, at room temperature. Data were analyzed by nanoparticle tracking analysis (NTA 3.2 software, Malvern) with detection threshold held constant between replicates. Polystyrene latex microsphere beads of 100 nm (Malvern) were used as a size standard. Data are shown as mean diameter ± standard deviation.

## Negative stain electron microscopy

Five-microliter samples were adsorbed for 1 min onto freshly glow-discharged Formvar-carbon-coated 400-mesh copper grids. The grids were washed with 0.1 M and 0.01 M ammonium acetate buffer (pH 7.4), stained with freshly filtered 1% uranyl acetate solution, and excess stain removed with filter paper. The grids were allowed to dry overnight and then viewed with a Tecnai F20 electron microscope (FEI Company / Thermo Fisher Scientific, Eindhoven, The Netherlands) at an acceleration voltage of 200 kV. Electron micrographs were recorded on an Eagle 4K charge-coupled device camera (FEI Company / Thermo Fisher Scientific, Eindhoven, The Netherlands).

## Primary hippocampal neuron assay

Primary hippocampal cultures were generated from E16 ~18 C57BL/6 mouse embryos. The hippocampi were dissected while immersed in Hank's Balanced Salt Solution buffered with 10 mM HEPES and cells were dissociated with 0.125% trypsin (Invitrogen Canada, Inc., Burlington, ON, Canada) for 10 ~ 15 min at 37°C, followed by trituration. Dissociated cells were plated at a density of 1.0 ~ 1.5 × 10$^5$ cells/cm$^2$ in 6-well plates pre-coated with poly-D lysine and grown in Neurobasal medium with B-27 supplement and Glutamax (Invitrogen). Half the medium was refreshed every 3

days. The treatment of the primary hippocampal neurons with $A\beta$ and other peptides was started on day 18. After treatment, cells were lysed in 0.5% SDS, 1% Nonidet P-40, 2 mM EDTA, 100 mM NaCl, 100 mM Tris, pH 7.6, supplemented with protease inhibitor and phosphatase inhibitor cocktails (Roche, Mississauga, ON, Canada) for Western blot analyses.

## Acknowledgements

This project received funding from the Canadian Institutes of Health Research (CIHR), a medical sciences proof-of-principle (MScPoP) funding program hosted by MaRS Innovation and the Ontario Centres of Excellence Inc., the Alberta Prion Research Institute (201600028), a Grant in Aid from the Heart and Stroke Foundation of Canada (G-15-0009148), and most generous private support from the Rosiak family. The mass spectrometry component of this study was enabled by generous philanthropic support from the Irwin family and a Leading Edge Fund (LEF) infrastructure grant from the Canada Foundation for Innovation (CFI). We acknowledge the Structural and Biophysical Core Facility, Hospital for Sick Children, for access to the NanoSight instrument. The electron microscopy facility was established with support from a New Initiatives Fund (NIF) award from the CFI.

## Additional information

### Competing interests

HWa, GS-U: Holds provisionary US patent on amyloid-beta binding polypeptides based on the results of this study (filing number 62/451,309). The other authors declare that no competing interests exist.

### Funding

| Funder | Grant reference number | Author |
|---|---|---|
| Canadian Institutes of Health Research | | Gerold Schmitt-Ulms |
| Ontario Centres of Excellence | | Simon Sharpe<br>Gerold Schmitt-Ulms |
| Alberta Innovates Bio Solutions | 201600028 | Holger Wille<br>Gerold Schmitt-Ulms |
| Heart and Stroke Foundation of Canada | G-15-0009148 | Simon Sharpe |
| Canada Foundation for Innovation | | Gerold Schmitt-Ulms |

The funders had no role in study design, data collection and interpretation, or the decision to submit the work for publication.

### Author contributions

HWa, Conceptualization, Data curation, Formal analysis, Validation, Investigation, Visualization, Methodology, Writing—review and editing; LDM, Data curation, Validation, Investigation, Methodology, Writing—review and editing; PG, Data curation, Investigation, Methodology, Writing—review and editing; DW, Data curation, Software, Investigation, Visualization, Writing—review and editing; MS, Investigation, Visualization, Writing—review and editing; AF, Formal analysis, Investigation, Methodology, Writing—review and editing; AR-R, Investigation, Methodology; RP, Conceptualization, Resources, Software, Supervision, Investigation; JCW, Conceptualization, Supervision, Methodology, Writing—review and editing; AC, Conceptualization, Formal analysis, Supervision, Methodology, Writing—review and editing; HWi, Conceptualization, Resources, Supervision, Investigation, Visualization, Methodology, Writing—review and editing; SS, Conceptualization, Resources, Data curation, Supervision, Funding acquisition, Methodology, Writing—review and editing; GS-U, Conceptualization, Resources, Data curation, Software, Formal analysis, Supervision, Funding acquisition, Validation, Investigation, Visualization, Methodology, Writing—original draft, Project administration, Writing—review and editing

## Author ORCIDs
Hansen Wang, http://orcid.org/0000-0003-4414-3372
Alejandro Ruiz-Riquelme, http://orcid.org/0000-0001-6581-7132
Régis Pomès, http://orcid.org/0000-0003-3068-9833
Holger Wille, http://orcid.org/0000-0001-5102-8706
Gerold Schmitt-Ulms, http://orcid.org/0000-0001-6962-0919

## Ethics

Animal experimentation: The work was performed in strict accordance with University of Toronto animal care and biosafety recommendations. All mice were handled according to procedures approved (AUP4183.3) by the animal care committee at University Health Network overseeing work at the Krembil Discovery Centre (Toronto). The handling of samples and reagents followed biosafety procedures approved (208-S06-2) by the University of Toronto Biosafety Program.

## Additional files

### Supplementary files
• Supplementary file 1. Subset of Aβ1-42 interactors sorted by protein category.

### Major datasets
The following dataset was generated:

| Author(s) | Year | Dataset title | Dataset URL | Database, license, and accessibility information |
| --- | --- | --- | --- | --- |
| Schmitt-Ulms U, Williams D | 2016 | The human amyloid beta peptide interactome | http://www.ebi.ac.uk/pride/archive/projects/PXD004867 | Publicly available at the EMBL-EBI PRIDE Archive (accession no: PXD004867) |

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
