## [Decision Letter]

[Editors’ note: a previous version of this study was rejected after peer review and an appeal, but the authors submitted for reconsideration. The first decision letter after peer review is shown below.]

Thank you for submitting your work entitled "The human Aβ interactome: binding to somatostatin favors formation of distinct oligomers and impairs detection of Aβ1-42" for consideration by *eLife*. Your article has been reviewed by three peer reviewers, one of whom is a member of our Board of Reviewing Editors, and the evaluation has been overseen by a Senior Editor. The following individuals involved in review of your submission have agreed to reveal their identity: Colin Masters (Reviewer #2).

Our decision has been reached after consultation between the reviewers. Based on these discussions and the individual reviews below, we regret to inform you that your work will not be considered further for publication in *eLife*.

Although the manuscript provides potentially useful information about Aβ interactors, after a thorough discussion among the editors and reviewers, we are particularly concerned that the physiologic basis of an effect of somatostatin14 on the aggregation of Aβ(1-42) has not been addressed in cells or in vivo. Further, the evidence for a role of somatistatin14 in the aggregation of Aβ(1-42) is not compelling.

Summarizing other issues that you will see in the full reviews appended below:

One implication of this work is that Aβ(1-42) can be found in the cytoplasm of nerve cells. What evidence is there that this is so? Are its termini free? What was the difference here between monomer and oligomer? Were there any fibrils? Did the authors use EM and aSEC? How do these structures relate to the recent papers by Wälti et al. and Colvin et al.?

The focus on somatostatin is explained, but this section is far too long. The authors must also discuss the effects of corticostatin on Aβ aggregation, given that this hormone was not identified as a positive interactor.

The issue of n numbers must be addressed. The number must be at least three, meaning extracts from three different brains. Much more detail must be given in relation to tissue preparation.

The authors must explain why the capture reaction was done at pH 8.0. How did this pH influence the interactors detected?

The denaturation and trypsinization steps do not seem to figure in Materials. Quite often, more details are needed, so that one could attempt to replicate these findings. The authors must go through the manuscript with this in mind. For instance, in subsection “Binding of SST14 or CST17 precludes detection of Aβ1-42 with commonly used antibodies”, much more detail is required about Western blotting using Aβ antibodies. At times, the authors refer to TMR-Aβ, at other times to Aβ-TMR. Does this mean the TMR was at varying ends?

Reviewer #1:

This is a comprehensive proteomic study. It provides useful background information, even though some of the interpretations might be open to question. The authors identified proteins within the human brain that monomeric and oligomeric Aβ(1-42) bind to. This work implies that oligomeric Aβ(1-42) is found in the cytoplasm. What is the evidence for this? How do the structures of monomeric and oligomeric Aβ(1-42) differ and how does this relate to the structures reported by Colvin et al. and Wälti et al.? What do the monomers and oligomers used here look like by EM? How do the authors know that there were no fibrils? Somatostatin was found to bind to the N-terminal region of Aβ(1-42) oligomers, but it is not listed in Table 1. The discussion relating to somatostatin was much too long. Overall, this manuscript is sophisticated in some parts, but it is rather unsophisticated in others. I could not understand what the n numbers were. Did the authors only use one frontal cortex (age of patient, post mortem delay, control or AD?). If so, the n number was one (which would be insufficient). Is it known if the post mortem delay could influence these interactions?

Reviewer #2:

This is a reasonable proteomic approach but the results are difficult interpret because of the experimental conditions used. Some would argue that monomeric Aβ does not occur in vivo, and that the c-terminus is rarely available for protein-protein interactions given its hydrophobic nature. Nevertheless the data are of use in future studies when other putative interactions are being evaluated. The focus on SST14 is a little surprising given that it did not come up in the primary screen.

Reviewer #3:

This is an interesting manuscript that describes a quantitative mass spec approach to investigate the Aβ interactome. Orthogonal approaches using N-terminal and C-terminal tags, and aggregated versus monomeric Aβ are appreciated and confirm several proteins previously implicated as Aβ binders. The data in Figure 1 and Figure 2 and associated tables will be of interest to many involved in Aβ research. By relaxing the requirement that peptides had to be at least six amino acids long to be considered for protein identification, the sequence "NFFWK" was identified. The fact that this sequence resides not simply in the preprosomatostatin and preprocortistatin, but also the mature hormones (SST14 and CST17) together with evidence of other SST14 sequence (TFTSC) justifies their focus on SST14. In Figure 3 the authors present evidence that SST14 can bind Aβ aggregates in vitro, and in Figure 4 they demonstrate that SST14 and CST17 can inhibit Aβ aggregation. However, CST17 is a much better inhibitor of Aβ aggregation, but given that this hormone was not identified as a positive interactor the relevance of this finding is unclear. The approach (SDS-PAGE and WB) used in Figure 5 and Figure 6 is weak and the conclusions drawn are shaky.

In terms of the various subsections of the manuscript, the Introduction was a joy to read, but the Results section was somewhat contorted and the Materials and methods lacked many details. Indeed, certain parts of the Materials and methods are incomplete rendering it impossible for independent replication. The Discussion contained some strong points, but was wandering and highly speculative. In sum, this manuscript contains several interesting and innovative points, but it is not suitable for publication in its current form.

A list of specific points documenting errors/inconsistencies and omissions is provided.

1) There are no details on how the brain extract was prepared. This is an important issue as the method used will influence the proteome in the extract. Similarly, there are no details regarding the donor(s) – i.e. cause of death, gender, age, etc. While the authors refer to 3 biological replicates, it is unclear if these means 3 extracts from the same brain, or extracts from 3 different brains. Clearly a proteomic analysis based on extracts from a single human brain would need validate using other human brains.

2) While the FRET experiments in Figure 3 are consistent with other data that support SST binding to Aβ aggregates, the experimental design of the experiments in Figure 3 preclude interpretation since they were done at pH close to the isoelectric point of Aβ. The apparent FRET and competition could result due to co-precipitation and formation of amorphous aggregates.

3) It is not clear why the capture reaction was done at pH 8.0, nor is there consideration of how this mildly alkaline pH may influence the interactors detected.

4) Subsection “Affinity capture of Aβ1-42 and its binding proteins in human frontal lobe extracts”: "the beads were additionally washed with 50 mL of 20 mM Hepes, pH 7.0, and transferred to Pierce Spin columns (catalog number 69705, Thermo Fisher Scientific, Inc.) to remove primary amines and prepare the samples for elution" – how were primary amines removed? Does this refer to depletion of the Tris in the lysis buffer?

5) Subsection “Workflow of Aβ interactome analyses”: "binders to the bait peptides were eluted by rapid acidification, fully denatured in 9 M urea, and trypsinized" – there are no details of the denaturation and trypsinization steps in the Results or Materials and methods.

6) Subsection “The Aβ1-42 interactome”: "Not surprisingly, the Aβ peptide itself was amongst the proteins identified whose enrichment levels in specific versus unspecific affinity capture samples were most pronounced" and Supplementary Table 1. Were the tryptic fragments detected restricted to Aβ, or as suggested on Supplementary Table 1, did they also include other regions of APP?

7) Subsection “Validation of SST14 binding to oAβ1-42”: "For this experiment, oAβ1-42 preparations (generated by incubating mAβ1-42 purified by gel filtration at 37 ºC for 2 h with shaking at the speed of 700 rpm)" – further details are required. Moreover, the material used should be characterized by EM and aSEC as it seems likely that the material used are not oligomers, but short fibrils. SDS-PAGE (Figure 3) is not a suitable means to determine native assembly size.

8) Subsection “FRET measurements”: "Solutions containing 20 μM concentrations of Edans-SST14, TMR-Aβ1-42 or a mixture of both fluorescently labeled peptides at pH 8.5 were rapidly adjusted to pH 5.2 and the resulting reaction mixtures incubated overnight". pH 5.2 is essentially the isoelectric point for Aβ, making interpretation of the presented result uncertain. TMR is hydrophobic and has a tendency to aggregate in aqueous solutions under conditions where the labeling density is sufficient to permit dye-dye interactions. A consequence of these interactions is fluorescence self-quenching.

9) At times the authors refer to TMR- Aβ, and at other times Aβ -TMR, they should clarify whether the TMR was on the N or C-terminus.

10) Subsection “Binding of SST14 or CST17 precludes detection of Aβ1-42 with commonly used antibodies”: "samples prepared as described in the previous section (but under omission of ThT) were analyzed by Western blotting using antibodies targeting N-terminal, central or C-terminal epitopes within Aβ1-42" – more details are necessary.

11) Figure 5 – the Ag stain data also show a decrease in the amount of peptide detected in the presence of SST14 or CST17. Moreover, truncation of the Ag stained gels at the 6 kDa marker makes it difficult to compare the Ag stained and WB samples.

12) Subsection “Aβ1-42 forms a distinct 50-60 kDa SDS-stable complex in the presence of SST14”: "To compensate for the abovementioned epitope masking effect of SST14 or CST17 (Figure 5), these experiments were undertaken with 5 to 10 times higher peptide concentrations than those described above (Figure 4 and Figure 5)." Why distort the experiment by using such high concentrations, why not simply use Ag staining since the authors claim that Ag stain was not influenced by the presence of SST14 or CST17.

13) Figure 6, lanes 1-4 are labeled "no SDS" – is this correct?

[Editors’ note: what now follows is the decision letter after the authors’ appeal.]

I apologize for the delay at all stages in the review of your work and I take responsibility for the decision to decline publication of this work. Because of our editorial process and the volume of submissions we receive (over 8000 last year), I do not become involved until the very end and then only in cases where the manuscript appears to be moving toward acceptance.

In the case of your submission, you are correct that the reviews contained suggestions that would have been possible for you to address, but on reading the comments of the referees in their individual critiques an in the ensuing consultation session, I was struck by the lack of any biological or even functional biochemical validation of your conclusion concerning the interaction of somatostatin and the Aβ peptide. I then conferred with two other *eLife* neuroscience Senior Editors who agreed that your results were simply too preliminary for one to conclude that the interaction has meaning in relation to Alzheimer's Disease. And in further consultation with the reviewers, two expressed concerns with the strength of your data regarding on influence of somatostatin on the aggregation of Aβ. However, they also were not in favor of demanding animal studies, which may be of dubious value given the inadequate animal models of Aβ aggregation and the development of dementia. (e.g. reviewer #3 - In my opinion, studies to show that the interaction between Aβ and SST14 has physiological or pathophysiologic significance is beyond the scope of this paper. Rather it is more important to understand if the experimental conditions in Figure 3 and Figure 4 really justify the conclusions. With regard to SST14's effect on Aβ aggregation, this is the least compelling part of the manuscript. I consider Figure 5 to be weak.)

So we are left with an unsatisfying conclusion that the solid proteomic data you have developed may or may not have any bearing on the physiologic role of somatostatin in the fate of extracellular Aβ peptide or oligomers. As a result, I felt we had no choice but to decline this work. However, if as you suggest it may be possible to evaluate the effect of somatostatin secreted by cells on the activity of Aβ monomer or oligomer in the phosphorylation of tau, then this at least could lead to a more plausible if not conclusive connection to disease. If you are willing to go one more round with us, I suggest you develop such an analysis and send it back to us for review by the same individuals who read the current version of your paper.

Sincerely, Randy Schekman

Editor-in-Chief

[Editors’ note: what now follows is the decision letter after the authors submitted for further consideration.]

Thank you for submitting your article "The human amyloid β peptide interactome: binding to somatostatin favors formation of distinct oligomers" for consideration by *eLife*. Your resubmission has been evaluated by a Senior Editor in consultation with the original reviewers. The editors would be happy to consider your work further once you have responded to new concerns raised by one of the original reviewers.

We remain somewhat conflicted about this submission. It has some strengths and the new data are a plus, but the physiological or more importantly pathophysiological role of the Aβ -SST interaction is still unclear.

The new data don't directly address the request I made in our exchange in your appeal letter:

"However, if as you suggest it may be possible to evaluate the effect of somatostatin secreted by cells on the activity of Aβ monomer or oligomer in the phosphorylation of tau, then this at least could lead to a more plausible if not conclusive connection to disease."

Instead, you used synthetic SST and synthetic Aβ. However, the new data in Figure 8 do indicate that co-addition of SST with Aβ monomer to primary neurons leads to increased phosphorylation of tau. So the authors have gone some way towards the request. Please comment on your use of synthetic peptides rather than those interacting in the medium of cultured cells.

The new data in Figure 7 demonstrate that Trp8 in SST is essential for the inhibition of Aβ aggregation, and that SST causes production of relatively small aggregates, this certainly bolsters the specificity and effect of SST, but this was considered a minor point.

---

## [Author Response]

[Editors’ note: the author responses to the first round of peer review follow.]

*Summarizing other issues that you will see in the full reviews appended below:*

*One implication of this work is that Aβ(1-42) can be found in the cytoplasm of nerve cells. What evidence is there that this is so?*

While there are in our view compelling prior reports by others that documented the presence of Aβ in mitochondria and the cytosol, including by super‐resolution imaging work [2], this is not something we emphasized in our report. Our choice to work with digitonin‐solubilized brain extracts during the discovery phase allowed us to keep an open mind in this regard, as these would include extracellular and most cellular proteins (except for nuclear proteins), consistent with the main subcellular areas previously reported to harbor Aβ. We then confirmed that Aβ can interact with a subset of previously reported intracellular Aβ interactors when they are offered in a brain extract (e.g., ATP synthase or ABAD). Importantly, for somatostatin, the most striking interactor revealed by our analysis, cytosolic uptake is not required for its Aβ interaction to occur, since both somatostatin and Aβ are secreted at the synapse.

*Are its termini free? What was the difference here between monomer and oligomer?*

We had tethered Aβ to affinity capture matrices using N‐terminal or C‐terminal biotin tags, which were coupled through a short linker (Experiments 1 and 2). We thereby generated separate interactome datasets for Aβ with free C‐ or N‐termini, which we reported in detail in the data tables. How oligomers were prepared was also described in our original submission.

*Were there any fibrils? Did the authors use EM and aSEC? How do these structures relate to the recent papers by Wälti et al. and Colvin et al.?*

Soluble oligomeric Aβ used for affinity capture were prepared using a previously published method (Goate et al., 1991 and Rogaev et al., 1985) known to generate amyloid‐β‐derived diffusible ligands (ADDLs). EM images characterizing these preparations have been published by others before but could also be included with a resubmission of our paper. The reviewer is correct in asserting that these included both oligomeric and prefibrillar Aβ. Larger aggregates would have sedimented during a 14,000 g centrifugation step, which we had applied prior to the capture of the Aβ bait.

*The focus on somatostatin is explained, but this section is far too long.*

This could easily be truncated.

*The authors must also discuss the effects of corticostatin on Aβ aggregation, given that this hormone was not identified as a positive interactor.*

One peptide we observed (NFFWK) could have originated from both cyclic hormones. A conclusion whether cortistatin was missing in the interactome dataset cannot be made. This is because, in contrast to somatostatin, no peptide was observed that uniquely identifies cortistatin. This observation could reflect one of several things, including the fact that the expression of cortistatin has been reported to be less abundant and more restricted in the brain, or could merely represent inadvertent ionization properties of the one tryptic peptide that is longer than 3 amino acids, which could uniquely identify cortistatin. Given that one peptide (TFTSC) uniquely identified somatostatin, it made sense to focus more on somatostatin during validation and in our discussion. The reviewer is correct in asserting that the effect of cortistatin on aggregation was even more pronounced than that observed for somatostatin. However, we would like to note that cortistatin did not behave in a qualitatively different manner from somatostatin in any of the assays.

Since November we have generated additional somatostatin‐Aβ interface mapping data, which inform about the minimum epitope required for binding and this could be included with a revised submission.

*The issue of n numbers must be addressed. The number must be at least three, meaning extracts from three different brains. Much more detail must be given in relation to tissue preparation.*

We are ourselves routinely frustrated by papers which report on interactome data that do not pass stringent conventions in the field. Our work is not one of them and describes three independent datasets, which together corroborated the conclusion that somatostatin was the most selective binder of oligomeric Aβ based on several peptide‐to‐spectrum matches and several biological and technical replicates (Experiments 1‐3). We did not anticipate that the robustness of the interactome data would be contentious and therefore did not mention that, in addition, we had previously generated a completely separate but almost identical Aβ interactome dataset. Samples in both studies were independently sourced, mass spectrometry analyses were undertaken on fundamentally different instruments (orbitrap versus triple time‐ of‐flight), one study was based on iTRAQ labeling, the other on TMT labels. Each time, brain extracts represented 1:1:1 (w/w/w) mixtures of frontal lobes from three elderly individuals that had died of non‐dementia causes. Each time, the NFFWK peptide derived from cortistatin/somatostatin was identified as the tryptic peptide that most selectively co‐purified with oligomeric Aβ but not monomeric Aβ. The datasets we included in the paper were produced on the cutting‐edge Orbitrap Fusion Tribrid mass spectrometer. If it was deemed helpful, we would be happy to also present the first complete dataset produced earlier on an AB Sciex 5600 machine when resubmitting our paper.

*The authors must explain why the capture reaction was done at pH 8.0. How did this pH influence the interactors detected?*

This pH may be the most often employed pH for interactome analyses, because it is applied in the widely used RIPA buffer formulation. Note that subsequent validation experiments were undertaken at physiological pH (ThT aggregation assays and gel based analyses) or lower pH (FRET analyses), demonstrating that the somatostatin‐Aβ interaction is tolerant of changes to pH.

*The denaturation and trypsinization steps do not seem to figure in Materials. Quite often, more details are needed, so that one could attempt to replicate these findings. The authors must go through the manuscript with this in mind. For instance, in subsection “Binding of SST14 or CST17 precludes detection of Aβ1-42 with commonly used antibodies”, much more detail is required about Western blotting using Aβ antibodies.*

We are cognizant of the frustrations that come with sloppily assembled Materials and methods sections. We are confident that you would have to agree that the respective section in our paper was assembled with care. Having stated this, we agree that it can be further improved and will be happy to add the requested details to clarify procedures further.

*At times, the authors refer to TMR-Aβ, at other times to Aβ-TMR. Does this mean the TMR was at varying ends?*

We apologize for this inaccuracy. The same reagent TMR‐derivative of Aβ was used in all experiments.

*Reviewer #1:*

*This is a comprehensive proteomic study. It provides useful background information, even though some of the interpretations might be open to question. The authors identified proteins within the human brain that monomeric and oligomeric Aβ (1-42) bind to. This work implies that oligomeric Aβ (1-42) is found in the cytoplasm. What is the evidence for this? How do the structures of monomeric and oligomeric Aβ (1-42) differ and how does this relate to the structures reported by Colvin et al. and Wälti et al.? What do the monomers and oligomers used here look like by EM?*

As mentioned above, this information could be included with a resubmission.

*How do the authors know that there were no fibrils?*

Only in so far as the Aβ we conjugated was soluble after 15 minute centrifugation at 14,000 g. The preparation method we used is expected to produce a highly heterogeneous ADDL mixture. This was intentional, as we did not want to bias the capture by focusing on a particular oligomer (e.g., those described by Ahmed et al., 2010 or Barghorn et al., 2005).

*Somatostatin was found to bind to the N-terminal region of Aβ (1-42) oligomers, but it is not listed in Table 1.*

Although somatostatin was most selectively co‐purifying with Aβ, it did not pass stringency criteria applied for proteins listed in Table 1, namely that identifications be based on a minimum of three peptides and that the lengths of those peptides exceed five amino acids. This point was included in the text but understandably can be missed and will be further emphasized by including a footnote to Table 1 in a revised document.

*The discussion relating to somatostatin was much too long.*

We will truncate this section.

*Overall, this manuscript is sophisticated in some parts, but it is rather unsophisticated in others. I could not understand what the n numbers were. Did the authors only use one frontal cortex (age of patient, post mortem delay, control or AD?). If so, the n number was one (which would be insufficient). Is it known if the post mortem delay could influence these interactions?*

See detailed response to this point above.

*Reviewer #2:*

*This is a reasonable proteomic approach but the results are difficult interpret because of the experimental conditions used. Some would argue that monomeric Aβ does not occur* in vivo*, and that the c-terminus is rarely available for protein-protein interactions given its hydrophobic nature.*

We agree with this comment. Note that we describe as a strict requirement for somatostatin binding to oligomeric Aβ that the latter is tethered through its C‐terminus and, therefore, has an accessible N‐terminus.

*Nevertheless the data are of use in future studies when other putative interactions are being evaluated. The focus on SST14 is a little surprising given that it did not come up in the primary screen.*

Not only did cortistatin/somatostatin come up in the screen but it was the most selective Aβ binder (based on the relative quantitation afforded by iTRAQ signature ions, when comparing samples and controls), once we waived the requirement that proteins had to be identified on the basis of three unique peptides of a minimum length of six amino acids. It therefore became the natural interactor to pursue further in validation work. This point will be further emphasized in a revised submission.

*Reviewer #3:*

*This is an interesting manuscript that describes a quantitative mass spec approach to investigate the Aβ interactome. Orthogonal approaches using N-terminal and C-terminal tags, and aggregated versus monomeric Aβ are appreciated and confirm several proteins previously implicated as Aβ binders. The data in Figure 1 and Figure 2 and associated tables will be of interest to many involved in Aβ research. By relaxing the requirement that peptides had to be at least six amino acids long to be considered for protein identification, the sequence "NFFWK" was identified. The fact that this sequence resides not simply in the preprosomatostatin and preprocortistatin, but also the mature hormones (SST14 and CST17) together with evidence of other SST14 sequence (TFTSC) justifies their focus on SST14. In Figure 3 the authors present evidence that SST14 can bind Aβ aggregates in vitro, and in Figure 4 they demonstrate that SST14 and CST17 can inhibit Aβ aggregation. However, CST17 is a much better inhibitor of Aβ aggregation, but given that this hormone was not identified as a positive interactor the relevance of this finding is unclear. The approach (SDS-PAGE and WB) used in Figure 5 and Figure 6 is weak and the conclusions drawn are shaky.*

These data were included because, in our view, in particular, the heterodimer bands shown in Figure 5 represented a powerful validation of the direct interaction between Aβ and SST. We also found it was worth sharing that oligomeric complexes observed in the presence of SST migrated at apparent molecular masses of 55‐65 kDa, which could only be revealed by SDS‐free gel‐based analyses. Finally, the gel‐based analyses documented that these somatostatin‐ dependent complexes can be observed with Aβ1‐42 but not Aβ1‐40, which would not have been apparent had we only shown the ThT assay data, in which both Aβ peptides behaved similarly

*In terms of the various subsections of the manuscript, the Introduction was a joy to read, but the Results section was somewhat contorted and the Materials and methods lacked many details. Indeed, certain parts of the Materials and methods are incomplete rendering it impossible for independent replication.*

We can make sure that a revised submission will not lack in this area.

*The Discussion contained some strong points, but was wandering and highly speculative.*

We will remove speculative points for a resubmission.

*In sum, this manuscript contains several interesting and innovative points, but it is not suitable for publication in its current form.*

*A list of specific points documenting errors/inconsistencies and omissions is provided.*

*1) There are no details on how the brain extract was prepared. This is an important issue as the method used will influence the proteome in the extract. Similarly, there are no details regarding the donor(s) – i.e. cause of death, gender, age, etc. While the authors refer to 3 biological replicates, it is unclear if these means 3 extracts from the same brain, or extracts from 3 different brains. Clearly a proteomic analysis based on extracts from a single human brain would need validate using other human brains.*

We apologize for this omission and regret to have left this point unclear. As mentioned in response to the editorial summary above, the study results were NOT based on an extract generated from a single human brain but represented highly robust data based on three completely separate interactome experiments (Experiments I ‐ III) undertaken with mixtures of frontal brain samples from several elderly individuals who died of non‐dementia causes. Note that due to the complexity of the human brain, these extracts were based on mixtures of myriads of different brain cells. As briefly mentioned above, the investigation of binders to oligomeric and monomeric Aβ was undertaken completely independently twice, each time with several biological replicates and controls (and we had decided to not include the first dataset undertaken as it did not add anything of substance and seemed excessive). We will improve on this aspect of the description and list the specifics of the frontal lobe brain samples used in the revised submission.

*2) While the FRET experiments in Figure 3 are consistent with other data that support SST binding to Aβ aggregates, the experimental design of the experiments in Figure 3 preclude interpretation since they were done at pH close to the isoelectric point of Aβ. The apparent FRET and competition could result due to co-precipitation and formation of amorphous aggregates.*

We had considered this point. We take it that the reviewer would agree that if quenching was merely due to unspecific formation of amorphous co‐precipitation of Aβ near its physiological pH (Figure 3), we would not have observed the acceptor sensitization at 596 nm and it would have been unlikely for the addition of unlabeled somatostatin, but not unlabeled AVP, to have rescued this quenching effect (see Figure 3).

*3) It is not clear why the capture reaction was done at pH 8.0, nor is there consideration of how this mildly alkaline pH may influence the interactors detected.*

Most co‐immunoprecipitation or affinity capture interactome analyses are undertaken at this pH due to the popularity of RIPA buffer in these applications.

*4) Subsection “Affinity capture of Aβ1-42 and its binding proteins in human frontal lobe extracts”: "the beads were additionally washed with 50 mL of 20 mM Hepes, pH 7.0, and transferred to Pierce Spin columns (catalog number 69705, Thermo Fisher Scientific, Inc.) to remove primary amines and prepare the samples for elution" – how were primary amines removed? Does this refer to depletion of the Tris in the lysis buffer?*

Yes, the reviewer is correct that the presence of Tris would preclude labeling with the isobaric labels we routinely use. We will make this clear in a revised submission.

*5) Subsection “Workflow of Aβ interactome analyses”: "binders to the bait peptides were eluted by rapid acidification, fully denatured in 9 M urea, and trypsinized" – there are no details of the denaturation and trypsinization steps in the Results or Materials and methods.*

We have used and published these procedures many times before (including in detailed Methods in Molecular Biology protocols) and usually refer to them by referencing prior articles. We will ensure to provide the detail requested in a resubmission.

*6) Subsection “The Aβ1-42 interactome”: "Not surprisingly, the Aβ peptide itself was amongst the proteins identified whose enrichment levels in specific versus unspecific affinity capture samples were most pronounced" and Supplementary Table 1. Were the tryptic fragments detected restricted to Aβ, or as suggested on Supplementary Table 1, did they also include other regions of APP?*

Although APP peptides that map to regions outside of Aβ were ‘detected’ by the algorithm, their assignments were low scoring (95% significance) and therefore not credible. In contrast, and not surprisingly, tryptic peptides derived from Aβ were identified with highest confidence and exhibited the relative TMT signature ion quantitation profiles expected from the design of the experiments.

*7) Subsection “Validation of SST14 binding to oAβ1-42”: "For this experiment, oAβ1-42 preparations (generated by incubating mAβ1-42 purified by gel filtration at 37 ºC for 2 h with shaking at the speed of 700 rpm)" – further details are required. Moreover, the material used should be characterized by EM and aSEC as it seems likely that the material used are not oligomers, but short fibrils. SDS-PAGE (Figure 3) is not a suitable means to determine native assembly size.*

EM data can be provided for these preparations together with a resubmission.

*8) Subsection “FRET measurements”: "Solutions containing 20 μM concentrations of Edans-SST14, TMR-Aβ1-42 or a mixture of both fluorescently labeled peptides at pH 8.5 were rapidly adjusted to pH 5.2 and the resulting reaction mixtures incubated overnight". pH 5.2 is essentially the isoelectric point for* Aβ*, making interpretation of the presented result uncertain. TMR is hydrophobic and has a tendency to aggregate in aqueous solutions under conditions where the labeling density is sufficient to permit dye-dye interactions. A consequence of these interactions is fluorescence self-quenching.*

The reviewer is correct and we are keenly aware of this confounder. We will be able to convincingly exclude this possibility.

*9) At times the authors refer to TMR-* Aβ*, and at other times* Aβ *-TMR, they should clarify whether the TMR was on the N or C-terminus.*

See above. This inconsistency in the data presentation in Figure 3 versus the Materials and methods section will be corrected.

*10) Subsection “Binding of SST14 or CST17 precludes detection of Aβ1-42 with commonly used antibodies”: "samples prepared as described in the previous section (but under omission of ThT) were analyzed by Western blotting using antibodies targeting N-terminal, central or C-terminal epitopes within Aβ1-42" – more details are necessary.*

We will add a more detailed description of how these samples were following ThT assay analysis (but without ThT added) dissolved in sample buffer and analyzed by gel electrophoresis.

*11) Figure 5 – the Ag stain data also show a decrease in the amount of peptide detected in the presence of SST14 or CST17. Moreover, truncation of the Ag stained gels at the 6 kDa marker makes it difficult to compare the Ag stained and WB samples.*

The area above 6 kDa did not reveal sample‐specific bands but was characterized by signals present in all samples at the same levels that are routinely observed in highly exposed silver stains. Because these signals were not derived from somatostatin, cortistatin or Aβ (as their intensities did not correlate at all with the differences in the amounts of these peptides in the samples) they did not aid in interpreting the binding biology that was the focus of the experiment. We therefore chose to cut them from the presentation but will present the full silver‐stained blot as supplemental information with the resubmission.

*12) Subsection “Aβ1-42 forms a distinct 50-60 kDa SDS-stable complex in the presence of SST14”: "To compensate for the abovementioned epitope masking effect of SST14 or CST17 (Figure 5), these experiments were undertaken with 5 to 10 times higher peptide concentrations than those described above (Figure 4 and Figure 5)." Why distort the experiment by using such high concentrations, why not simply use Ag staining since the authors claim that Ag stain was not influenced by the presence of SST14 or CST17.*

The silver‐staining was not sensitive enough to detect intermediate‐sized bands, which we interpreted to consist of Aβ and somatostatin, and which were detectable with 6E10.

13) Figure 6, lanes 1-4 are labeled "no SDS" – is this correct?

Yes, these samples were generated under omission of SDS. We will provide a detailed description of the running conditions of the gel when resubmitting the paper.

[1] Jin M, Shepardson N, Yang T, Chen G, Walsh D, Selkoe DJ. Soluble amyloid beta-protein dimers isolated from Alzheimer cortex directly induce Tau hyperphosphorylation and neuritic degeneration. Proc. Natl Acad Sci USA. 2011; 108(14):5819-24

[2] Kaminski Schierle GS, van de Linde S, Erdelyi M, Esbjorner EK, Klein T, Rees E, et al. In situ measurements of the formation and morphology of intracellular beta‐amyloid fibrils by super-resolution fluorescence imaging. J AM Chem. 2011; 133(33):12902-5. [3] De Felice FG, Velasco PT, Lambert MP, Viola K, Fernandez SJ, Ferreira ST, et al. Aβ oligomers induce neuronal oxidative stress through an N‐methyl‐D‐aspartate receptor‐dependent mechanism that is blocked by the Alzheimer drug memantine. J Biol Chem. 2007;282(15):11590‐601.

[4] Krafft GA, Klein WL. ADDLs and the signaling web that leads to Alzheimer's disease. Neuropharmacology. 2010;59(4‐5):230‐42.

[5] Ahmed M, Davis J, Aucoin D, Sato T, Ahuja S, Aimoto S, et al. Structural conversion of neurotoxic amyloid‐beta(1‐42) oligomers to fibrils. Nat Struct Mol Biol. 2010;17(5):561‐7.

[6] Barghorn S, Nimmrich V, Striebinger A, Krantz C, Keller P, Janson B, et al. Globular amyloid beta‐peptide oligomer ‐ a homogenous and stable neuropathological protein in Alzheimer's disease. J Neurochem. 2005;95(3):834‐47.

[Editors’ note: the authors’ response to the appeal decision letter follows]:

*[…] If you are willing to go one more round with us, I suggest you develop such an analysis and send it back to us for review by the same individuals who read the current version of your paper.*

*Sincerely, Randy Schekman*

*Editor-in-Chief*

Thank you again for your thoughtful comments and for having provided us with the opportunity to resubmit a revised manuscript. Over the past few weeks, we have reworked the manuscript and have added several new analyses to strengthen its conclusions.

1. New primary hippocampal neuron assay data (Figure 8), which assessed if the presence of monomeric or pre‐aggregated somatostatin (SST) affects Aβ‐dependent tau hyperphosphorylation.

You will find that this analysis uncovered a surprising effect of SST on Aβ‐dependent tau hyperphosphorylation. Normally, Aβ only triggers tau hyperphosphorylation in this assay (first described in detail by Dennis Selkoe’s group [7]) after it has been subjected to a pre‐aggregation step, i.e., tau hyperphosphorylation is never observed following addition of a monomeric preparation of Aβ to the cell culture medium. Our new data document that the concomitant addition of pre‐aggregated SST and monomeric Aβ can induce tau hyperphosphorylation. Importantly, neither monomeric nor pre‐aggregated SST alone promotes tau hyperphosphorylation, validating that the tau hyperphosphorylation effect of pre‐aggregated SST depends on monomeric Aβ being also added to the cell culture dish. Moreover, consistent with our other in vitro data, which revealed that SST (and its paralog cortistatin) interact with Aβ, but not other cyclic neuropeptides (AVP, oxytocin or bombesin), we document that the addition of pre‐aggregated preparations of these negative control peptides and monomeric Aβ to the cell culture medium does not induce tau hyperphosphorylation.

2. New epitope mapping data that define elements within SST responsible for binding to Aβ (Figure 7).

This experiment was undertaken in response to reviewer requests to strengthen SST‐Aβ interaction data presented in Figure 3 and 4. The new results demonstrate that the SST‐Aβ interaction depends on a specific SST residue. More specifically, replacement of tryptophan‐8 within SST with another amino acid abolished the lag phase extension observed in the Aβ aggregation assay in the presence of SST. Moreover, other new data, generated with truncated SST peptides, indicate that the stretch of SST residues 5‐11 surrounding the critical tryptophan‐8 is sufficient to induce the lag phase extension. In other words, consistent with data we had included in our original submission, which documented that the lag phase extension also occurred when the disulfide bridge within SST was reduced, these new results corroborate the conclusion that binding of SST to Aβ does not rely on the SST being present in its cyclic form. Notably, the SST epitope for binding to Aβ comprises the previously known minimal epitope (phenylalanine‐7‐tryptophan‐8‐lysine‐9‐threonine‐10) required for SST’s biological activity, which depends on its binding to one of five known SST receptors (SSTR1‐5)[8]. This observation is interesting as it raises the prospect that Aβ may compete with and thereby modulate SST ligand‐receptor interactions.

3. New nanoparticle tracking analysis that explores particle size distributions of Aβ in the presence or absence of SST (Figure 7—figure supplement 1)

This experiment was conducted in response to reviewer criticism of Figure 5 and Figure 6 of our original submission, which documented differences in Aβ‐specific signals when ThT aggregation assay fractions, generated in the presence or absence of SST, were analyzed by Western blotting. To determine if these differences in Aβ assemblies were merely artifacts of sample handling, i.e., introduced by denaturing conditions applied during Western blot analysis, we sought to compare Aβ aggregate size populations in the presence or absence of SST by an orthogonal method that would not be expected to suffer from Western blot confounders. Nanoparticle tracking analysis (based on Nanosight technology [9]) is such a method that can be used to characterize complex populations of molecules of differing particle size. The new data document that Aβ aggregates present in ThT aggregation assay fractions after 2 hours of incubation are profoundly shifted toward smaller particle sizes when samples were co‐incubated with SST. Importantly, the replacement of SST with an SST‐W8P negative control peptide did not result in this particle size shift toward smaller aggregates, indicating that the altered particle size distribution is caused by the SST‐Aβ interaction, rather than by some unspecific effect of the peptide.

4. Inclusion of full‐view silver‐stained gels as a figure supplement (Figure 5—figure supplement 1).

One reviewer expressed concern about the fact that we had truncated the silver‐stained images depicted in Figure 5 and C. We have now included the full view of these gels. As described in our previous appeal letter, these gels document that the respective Aβ and SST (CST) bands, which we had highlighted in our original submission were the only bands visible in the low mass range of the gel. Other signals visible on the gel in the higher mass range did not follow the logic of the experiment and, therefore, were interpreted as artefacts, often seen in overexposed silver‐stains of this kind. Importantly, the levels of these high mass signals indicate no irregularities in the loading and running that by themselves could explain the differences in bands observed in the low mass region.

5. New electron microscopy data that compare Aβ assemblies in the presence or absence of SST (Figure 7—figure supplement 2).

We followed the request to include electron micrographs that depict how the presence of SST affects visible structures. As expected, Aβ1‐42 that was incubated in PBS resulted in the formation of characteristic amyloid fibrils, while Aβ1‐42 that was incubated with equimolar concentrations of SST formed small oligomeric assemblies instead. This observation confirmed the effects SST is having on the quaternary structure of Aβ.

6. Revised Materials and Methods section.

The revised paper contains a considerably expanded Materials and Methods section. Not only were original descriptions of methodologies improved but several new experimental methods are included that describe methodologies for the primary hippocampal neuron assay, the epitope mapping analyses, the nanoparticle tracking analyses and the electron microscopy work.

7. Revised Discussion and text in other sections.

We followed the suggestion of reviewers to cut down the Discussion. In the revised document, the speculative section dedicated to how data presented in this study may lead to a revised model of the role of SST in Alzheimer’s disease has been removed. However, we have strengthened the critical evaluation of our data. We also addressed reviewer comments by inserting in several places in the manuscript additional information that clarifies aspects of our work that could easily be missed or were not well explained in the first submission.

In conclusion, we again thank all reviewers for constructive comments and for taking the time out of their busy schedules to review our manuscript. We are also most appreciative of the considerable efforts invested by the editorial team of *eLife* to improve our submission and hope you will agree that the manuscript is now ready for publication.

References:

[7] Jin M, Shepardson N, Yang T, Chen G, Walsh D, Selkoe DJ. Soluble amyloid beta‐protein dimers isolated from Alzheimer cortex directly induce Tau hyperphosphorylation and neuritic degeneration. Proc Natl Acad Sci U S A. 2011;108(14):5819‐24.

[8] Pohl E, Heine A, Sheldrick GM, Dauter Z, Wilson KS, Kallen J, et al. Structure of octreotide, a somatostatin analogue. Acta Crystallogr D Biol Crystallogr. 1995;51(Pt 1):48‐59.

[9] Filipe V, Hawe A, Jiskoot W. Critical evaluation of Nanoparticle Tracking Analysis (NTA) by NanoSight for the measurement of nanoparticles and protein aggregates. Pharmaceutical research. 2010;27(5):796‐810.

[Editors’ note: the author responses to the re-review follow.]

*We remain somewhat conflicted about this submission. It has some strengths and the new data are a plus, but the physiological or more importantly pathophysiological role of the* Aβ *-SST interaction is still unclear.*

In our view, the pathophysiological role of the Aβ ‐SST interaction will only come into full view, once we learn if and how the presence or absence of SST (and possibly CST) affects pathological hallmarks of Alzheimer’s disease in mice and humans. The experiments in primary hippocampal neurons provide a first hint but admittedly fall short of providing conclusive answers. While we would have loved to be able to address also this point more conclusively, in our view the in‐depth Aβ interactome data we provided, together with the discovery and detailed characterization of the Aβ ‐SST interaction, represents a major contribution in itself.

*The new data don't directly address the request I made in our exchange n your appeal letter:*

*"However, if as you suggest it may be possible to evaluate the effect of somatostatin secreted by cells on the activity of Aβ monomer or oligomer in the phosphorylation of tau, then this at least could lead to a more plausible if not conclusive connection to disease."*

*Instead, you used synthetic SST and synthetic* Aβ.

The perceived deviation in the design of the primary hippocampal neuron assay data was caused by a misunderstanding. When we wrote the appeal letter, we did not intend our planned experiments to be based on ‘secreted’ SST (please see below for explanations) but had in mind the very analysis we included in the resubmission. More specifically, the statement we used in our original appeal letter was: “Less conclusive, yet doable experiments, in which we will investigate if the presence of somatostatin in the cell culture medium influences effects of monomeric or oligomeric Aβ preparations on tau phosphorylation in primary hippocampal neurons (analogous to published work by Jin et al., 2011), were initiated immediately following our *eLife* submission and, hopefully, will lead to results within the next month.”

*However, the new data in Figure 8 do indicate that co-addition of SST with* Aβ *monomer to primary neurons leads to increased phosphorylation of tau. So the authors have gone some way towards the request. Please comment on your use of synthetic peptides rather than those interacting in the medium of cultured cells.*

There are several considerations that guided our choice to base the experiment on the addition of synthetic Aβ and synthetic SST:

1) Early in its biogenesis SST is directed toward the regulated secretory pathway. Several conditions have been shown by others to induce or modulate the regulated release of neuropeptides, including SST (e.g., depolarization of the cellular membrane followed by Ca^2+^ influx through voltage‐sensitive Ca^2+^ channels (Kang et al, 1987), or addition of growth hormone, dopamine, GABA, acetylcholine, or a subset of so‐called small‐molecule secretagogues). However, the timed release of a considerable proportion of SST stores from neurons in a way that does not perturb other aspects of the cellular homeostasis, including kinase signaling, represents a challenge. In the absence of a robust method that causes the selective release of SST, the cell culture assay would have needed to be based on steadystate levels of this peptide in the cell culture medium. For generating a negative control, this would have forced a need to inhibit SST release, to sequester SST from the medium (so it cannot interact with Aβ), or to work with cells obtained from SST knockout mice. While we have obtained an SST knockout mouse model and are currently undertaking cross‐breeding experiments with AD model mice, our breeding colony is too small for dedicating mice at this time to experiments we consider high risk (see below). Part of this risk stems from our expectation that kinase signaling pathways might be altered in cells from these mice, due to previously established influences of SST receptor activation on kinase signaling pathways (Goate et al., 1991; Rogaev et al., 1995; Sherrinton et al. 1995 and Yang, Li, Walsh and Selkoe, 2017).

2)Note that the release of SST into the cell culture medium would still be a poor mimic of the natural release of this peptide into the extracellular space within the architecture of the brain. This is because it is likely that in the cell culture dish SST would be more rapidly dispersed and its levels diluted upon its release into the relatively excessive volume of the cell culture medium. In contrast, in the brain, where cells exist in a relatively ‘dry’ environment, even modest levels of SST release may lead to localized spikes in its concentration.

3) In fact, because SST is stored in dense core granules as amyloid‐like aggregates (Selkoe, 20010, it is to be expected that in proximity to its natural release sites high concentrations of this peptide, including residual aggregates of varying size, are present in the brain. The disintegration of amyloid aggregates released in this manner has been observed to play out in time‐scales of several hours (Narayan et al., 2012). Thus, it is plausible that the low μM concentrations of synthetic SST, which we chose for our experiments, can also be encountered in the brain. Note also that there is no evidence that secreted or synthetic SST are different in their composition or post‐translational modifications.

4)In the brain, SST is predominantly released by GABAergic interneurons. While it is conceivable that one may enrich these by FACS sorting, such methods would be expected to return a dissatisfactory cell yield, below the number of cells required for carrying out the hippocampal neuron assay. Note that even in its current form, one female mouse that is pregnant with an average number of embryos allows us to test no more than 12 different assay conditions. Because interneurons represent a small fraction of the total hippocampal cell count, their FACS selection would push this assay to its limit (or beyond it), certainly not allowing a side‐by‐side comparison of the exposure of cells harvested from littermates to different peptide mixtures.

5)The cell culture assay that monitors tau hyperphosphorylation in response to the addition of synthetic Aβ was optimized and described in detail for primary hippocampal mouse neurons harvested on day 16‐18 from embryos and cultured in a specific cell culture medium for an additional 18 days. Any deviation from this protocol would have exposed experiments to criticism that the data cannot be compared to preexisting data generated with this assay. While this may seem a minor consideration, the Alzheimer’s disease research community generally dismisses cell‐based assay results as not being reflective of signaling underlying disease. The assay format we employed is, to our knowledge, the only cell culture based assay that enjoys relatively broad acceptance in the field for experiments probing Aβ ‐dependent tau hyperphosphorylation, due to data by others, which established similar tau phosphorylation site profiles observed in the disease and in analyses based on this assay.

6) The addition of synthetic peptides allowed us to generate several informative controls. Note that had we been able to overcome some of the aforementioned hurdles and work with endogenously released SST, we still would not have been able to compare the effect of SST (or CST) with those of other cyclic neuropeptides, including AVP and oxytocin, or the peptide bombesin, which is not cyclic but has been described to share with SST similar aggregation characteristics (see Table S1 in Selkoe, 2001). Finally, the assay design we selected allowed us to add SST to the cell culture medium in monomeric or pre‐aggregated form, thereby providing the insight that for the SST‐dependent tau hyperphosphorylation to be observed, cells had to be exposed to pre‐aggregated SST AND monomeric Aβ. This mixture is remarkable in that it strictly relied on monomeric Aβ also being present, yet we have never observed monomeric Aβ by itself to cause tau hyperphosphorylation.

A few sentences have been added to the manuscript to explain some of these considerations to the interested reader.

*The new data in Figure 7 demonstrate that Trp8 in SST is essential for the inhibition of* Aβ *aggregation, and that SST causes production of relatively small aggregates, this certainly bolsters the specificity and effect of SST, but this was considered a minor point.*

Agreed.